# The transformation of the forest steppe in the lower Danube Plain of south-eastern Europe: 6000 years of vegetation and land use dynamics

Angelica Feurdean[1,2,3]*#, Roxana Grindean[3] #, Gabriela Florescu[3,4,5], Ioan Tanţău[3], Eva Niedermeyer[2],
Andrei-Cosmin Diaconu[3], Simon M Hutchinson[6], Anne Brigitte Nielsen[7], Tiberiu Sava[8], Andrei Panait[2],
Mihaly Braun[9], Thomas Hickler[1,2]

[1]Department of Physical Geography, Goethe University, Altenhöferallee 1, 60438 Frankfurt am Main, Germany

[2]Senckenberg Biodiversity and Climate Research Centre (BiK-F), Senckenberganlage, 25, 60325, Frankfurt am Main, Germany

[3]Department of Geology, Babeș-Bolyai University, Kogălniceanu 1, 400084, Cluj-Napoca, Romania

[4]Department of Geography, Stefan cel Mare University, 13 Universităţii Street, 720229, Suceava, Romania

[5]Department of Botany, Faculty of Science, Charles University, CZ-128 01 Prague, Czech Republic

[6]School of Science, Engineering and Environment, University of Salford, Salford, M5 4WT, UK

[7]Department of Geology, Lund University, Sölvegatan 12, 22362 Lund, Sweden

[8]Horia Hulubei National Institute for Physics and Nuclear Engineering (IFIN-HH), Reactorului 30, 077125, Măgurele, Romania

[9]Institute for Nuclear Research of the Hungarian Academy of Sciences, H-4026 Debrecen, Bem tér 18/C, Debrecen, Hungary

*Correspondence to*: Angelica Feurdean (Feurdean@em.uni-frankfurt.de)
 # These authors contributed equally to this work

**Abstract**

Forest steppes are dynamic ecosystems, highly susceptible to changes in climate, disturbances, and land use. Here we examine the Holocene history of the European forest steppe ecotone in the Lower Danube Plain to better understand its sensitivity to climate fluctuations, fire and human impact, and the timing of its transition into a cultural forest steppe. We used multi-proxy analyses (pollen, *n*-alkanes, coprophilous fungi, charcoal and geochemistry) of a 6000-year sequence from Lake Oltina (SE Romania) combined with a REVEALS model of quantitative vegetation cover. We found a greater tree cover, composed of

xerothermic (*Carpinus orientalis* and *Quercus*) and temperate (*Carpinus betulus*, *Tilia, Ulmus* and *Fraxinus*) tree taxa, between 6000 and 2500 cal yr BP. Maximum tree cover (~50%) dominated by *C. orientalis* occurred between 4200 and 2500 cal yr BP at a time of wetter climatic conditions and moderate fire activity. Compared to other European forest steppe areas, the dominance of *C. orientalis* represents the most distinct feature of the woodland's composition during that time. Tree loss was under way by 2500 yr BP (Iron Age) with the REVEALS model indicating a fall to ~20% tree cover from the late Holocene forest maximum linked to clearance for agriculture, while climate conditions remained wet. Biomass burning increased markedly at 2500 cal yr BP suggesting that fire was regularly used as a management tool until 1000 cal yr BP when woody vegetation became scarce. A sparse tree cover, with only weak signs of forest recovery, then became a permanent characteristic of the Lower Danube Plain highlighting more or less continuous anthropogenic pressure. The timing of anthropogenic ecosystem transformation here (2500 cal yr BP) falls in between that in central eastern (between 3700 and 3000 cal yr BP) and eastern (after 2000 cal yr BP) Europe. Our study is the first quantitative land cover estimate at the forest steppe ecotone in south eastern Europe spanning 6000 years and provides critical empirical evidence that, at a broad spatial scale, the present-day forest steppe /woodlands reflect the potential natural vegetation in this region under current climate conditions. However, tree cover extent and composition have neither been stable in time nor shaped solely by the climate, highlighting the need to consider vegetation in a dynamic way under changing environmental conditions (climate, natural disturbances and human impact).

## 1.    Introduction

Projected changes in climate and increasing human environmental impacts are generating global concern about the functioning of ecosystems as well as the provision of ecosystem services (IPCC, 2014; IPBES, 2019). Lowland ecosystems (mesic and steppic grasslands, woodlands, etc) provide an array of provisioning (e.g. crops, grazed areas, wood) and regulating services (e.g. soil protection; European Environmental Agency, 2016). However, in comparison to the mountainous areas of central eastern Europe, lowland ecosystems, especially steppe grasslands, have been more strongly impacted by human activities (Magyari et al., 2010; Tonkov et al., 2014; Feurdean et al., 2015; Kuneš et al., 2015; Novenko et al., 2016; Shumilovkiksh et al., 2018, 2019; Jamrichová et al., 2019; Vinze et al., 2019; Clearly et al., 2019; Gumnior et al., 2020). This partly reflects the lowlands' deeper and more fertile soils and the greater accessibility of the terrain, which have promoted extensive agro-pastoral activities and human settlement. Lowlands also include more frequent ecotones i.e., woodland/grassland borders, which are naturally more sensitive to climate change (Bohn et al., 2003).

According to Bohn et al. (2003), the potential natural vegetation (PNV) in the study area, the easternmost part of the Lower Danube Plain, also known as the Southern Dobrogea Plateau, is forest steppe, i.e., woodland patches within a matrix of graminoid and forb dominated communities. It borders steppe grasslands i.e., treeless vegetation dominated by graminoids and forbs to the east. The forest steppe and steppe vegetation extend over 6000 km along an east-west gradient across Eurasia (Fig.

1) under climate conditions delimited by a ca. 2 month long late-summer drought for the forest steppe zone and 4–6 months for the steppe (Walter, 1974). However, according to the management plan for the region (Planul de Management, 2016), there are currently few patches of natural steppe vegetation preserved in the Southern Dobrogea Plateau as most have become ruderal steppe. Since the Lower Danube Plain represents one of the oldest areas of continuous human occupation from Neolithic onwards i.e., 8000 cal yr BP (http://ran.cimec.ro; Bălășescu and Radu, 2004; Weininger et al., 2009; Wunderlich et al., 2012; Nowacki et al., 2019: Preoteasa et al., 2019) and one of the most important agricultural areas in Europe (European Environmental Agency, 2016), the current vegetation is likely very different from its natural state, though exactly how different is not known. The use of PNV as a baseline for natural vegetation has been found to be problematic as it considers only climax vegetation i.e., the final stage of an ecological succession, and assumes that vegetation remains static in space and time (Chiarruci, et al., 2010; Jackson et al., 2013; Abraham et al., 2016; Rull 2015). PNV also fails to consider the vital role of natural disturbances such as fire and herbivores and their impacts on vegetation succession (Chiarruci, et al., 2010; Feurdean et al., 2018), leading to unrealistic vegetation reconstruction even in the absence of disturbance by humans (Jackson et al., 2013). As a consequence, many areas that are currently covered by grasslands or open woodlands in Central Eastern Europe are defined as naturally dominated by deciduous broadleaf forest or mixed coniferous and broadleaf forest as the PNV (Feurdean et al., 2018) and the representation of pioneer trees is much lower compared to pollen-based Holocene vegetation estimates (Abraham et al., 2016). Inaccurate identification of natural vegetation types can also lead to inappropriate decision making in terms of conservation practices and policies. For example, the Global Partnership on Forest and Landscape Restoration use PNV to identify opportunities for landscape restoration to mitigate climate change in areas where the climate can sustain forest (http://www.wri.org/applications/maps/flr-atlas). However, this approach threatens grassland ecosystems as such policies are based on the false assumption that most grasslands are man-made and ignores the very high richness of grasslands at smaller spatial scale (Whittaker et al. 2001), as well as their unique cultural significance (Dengler et al., 2014).

Palaeoecological records provide a way to assess the past natural vegetation of a region and the legacy of natural disturbances and anthropogenic impacts on the landscapes (Willis and Birks, 2006). However, due to the dry climate of the Lower Danube Plain, very few palaeoecological archives are available to document past natural vegetation types. This leaves many open questions on the temporal dynamics of vegetation composition and drivers of changes in this region, and how its vegetation composition compares to other forest steppe areas in central-eastern Europe (Magyari et al., 2010; Feurdean et al., 2015; Kuneš et al., 2015), south-eastern Europe (Tonkov et al., 2014; Marinova 2006) and the Eastern European Plain (Novenko et al., 2016; Shumilovkiksh et al., 2018). Most of the archaeological and loess deposits in the region are devoid of an absolute chronology, have poor lithological context and lack favourable pollen preservation (Tomescu 2000). Nevertheless, based on these fragmentary records, it appears that steppe may have covered the landscapes of this region in the early Holocene, while forest-steppe vegetation may have expanded during a moist phase of the mid Holocene (Feurdean et al., 2014; Tomescu et al., 2004; Wunderlich et al., 2012; Hansen et al., 2015). In addition, models of deforestation rates, using a scenario that accounts for population history and technological advances, suggest that the extent of deforestation in the lower Danube basin has increased continuously since 4000 cal yr BP (Kaplan et al., 2009; Giosan et al., 2012). Indirect evidence for the Holocene

persistence of steppe grasslands in this region comes from the genetic investigation of steppe species including *Adonis vernalis*, *Astragalus exscapu*s, *Stipa capillata*, and *S. pulcherrim*, which mostly show a unique genetic diversity reflecting the continuous occurrence and limited past migration of these steppe elements within Europe (Kropf et al., 2020; Plenk et al., 2020).

Here, we explore the long-term vegetation dynamics of the Lower Danube Plain landscapes and the competing driving forces (climate, fire and anthropogenic impact). More specifically, we address the following research questions:

     i)       Is forest steppe the natural vegetation type of the Lower Danube Plain under climatic conditions similar to those in the present?

     ii)      Has the tree cover been more extensive or dominated by other tree taxa in the past?

     iii)     When did this area undergo the most marked land cover and land use changes and was this transformation continuous in time?

This study is built on a pollen-based quantitative vegetation reconstruction (REVEALS model), along with records of long-chain higher plant-wax *n*-alkanes, charcoal, coprophilous fungi and geochemistry from a sedimentary sequence from Lake Oltina, south-eastern Romania. This is the first pollen-based quantitative land cover estimate in the forest steppe of south eastern Europe. It allows the hypothesis of the naturalness of forest steppe ecosystems in this region, as well as its sensitivity to climate and anthropogenic impact, to be critically tested.

## 2 Study area

### 2.1 Geography, climate and vegetation

Lake Oltina (44°09'16"N 27°38'13"E, 7 m a.sl.) is located on the floodplain of the Danube river, the Lower Danubian Plain, in south-eastern Romania (Fig. 1). It is the largest fluvial lake on the Danube floodplain with a surface area of 33 km$^2$ and is part of a Natural Reserve and a NATURA 2000 site (ROSPA0056; Management Plan 2016). The lake has a major tributary, Caranaua Fetii, and is connected to two smaller lakes, Ceamurlia and Iortmac, located upstream as well as to the Danube river via an artificial dam built to prevent flooding (Telteu, 2014). The climate in the study region is wet-warm temperate continental (Koppen-Geigger class Dfa), also termed excessive with the prevalence of harsh winters and hot summers (Posea et al., 2005), due to the influence of air masses from continental Asia. The mean annual temperature is of ca. 11°C, mean January temperature of ca. -1 °C and a mean summer temperature 25°C. Annual precipitation is about 400 mm (Adamclisi Meteo station). The geology of the catchment comprises limestone and loess deposits, whereas the soils are represented mainly by haplic and luvic chernozioms and phaenozems (IUSS WRB 2006). Friable steep loess deposits are frequently eroded, delivering a clastic sedimentary influx into the lake (Romanescu et al., 2013).

The main potential vegetation types of the region include forest, forest steppe and calciphile steppe (Bohn at al., 2003). The main forest types are: thermophilous mixed deciduous broadleaf forests (subtypes G21, G22 and G34 according to Bohn et al. 2003); mesophytic deciduous broadleaf forests (subtypes F49, F67); forest steppe (subtype L13) and steppe (subtype M5). Additionally, alluvial forests (subtypes U19, U20), halophytic vegetation (P33) and tall reed vegetation and sedge swamps

(R1) prevail (Bohn et al., 2003). Corine Land Cover data (2012) indicates that the present land cover within a 20 km radius of the lake comprises ca. 65% arable land and orchards, 19% steppe and semi-natural grassland and 16% deciduous forest (S1; Grindean et al., 2019). Important tree species in forests within this radius, and thus the most relevant for the pollen source area, are *Quercus pubescens*, *Q. pedunculiflora*, *Q. robur*, *Q. cerris*, *Q. virgiliana*, *Carpinus orientalis*, *Acer tataricum*, *Tilia tomentosa*, *Fraxinus excelsior*, *Ulmus minor*, whereas hygrophilous tree taxa, growing along the Danube River, are represented

by *Populus nigra*, *P. alba*, *Salix alba* and *S. fragilis*. Shrubs occur abundantly either as understory vegetation or as thickets and are composed of *Fraxinus ornus*, *Cotinus coggygria*, *Prunus mahaleb*, *P. spinosa*, *P. cerasifera*, *Rosa canina*, *Pyrus communis*, *Crataegus monogyna*, *Amorpha fruticosa*, *Gleditsia triacanthos*, *Elaeagnus angustifolia* and *Ailanthus altissima.* Natural grassland steppic species are common on calcareous slopes and include *Stipa stenophylla*, *S. ucrainica*, *S. capillata*, *Poa angustifolia*, *Festuca valesiaca*, *Botriochloa ischaemum*, *Artemisia austriaca*, *Agrostis tenuis*, *Carex humilis*, *Centaurea*

*orientalis*, *Astragalus ponticus* and *Thymus marschallianus* (Sârbu et al., 2009; Grindean et al., 2019).

## 3 Material and Methods

### 3.1 Core collection, lithology and chronology

Sediment cores were extracted with a Livingstone piston corer (1m long, 5cm diameter) from the central part of the lake (1.8m

water depth) in spring 2016. The less consolidated sediment at the surface (36cm) had previously been retrieved with a gravity corer in 2014. A lithostratigraphic description was made according to changes in texture, colour, magnetic susceptibility and the organic carbon content (LOI).

Volume magnetic susceptibility ($\kappa_{vol}$) was measured with a Bartington Instruments Ltd MS2 meter and E point sensor (Bartington Instruments, 2008). For LOI, samples were dried overnight at 105 °C, combusted for 5 hours at 550°C and then

for 2 hours at 900°C, and LOI is expressed as percentage loss of the dry weight (Heiri et al., 2001). A composite sedimentary core record totalling 964 cm was constructed using the uppermost 10 cm from the gravity core and ten overlapping Livingstone core sections. To determine the erosion and anoxic conditions in the lake, elemental geochemical concentration was measured on 190 sediment samples (extracted at ~5 cm intervals along the core) and subsequently dried and homogenised using a non-destructive Niton XL3t 900 X-Ray Fluorescence analyser (fpXRF). NCS DC73308 was employed as a Certified Reference

Material (CRM). Measurement follows the procedure described by Hutchinson et al. (2015). We selected detrital element Zr as proxy for erosion (Kylander et al., 2011) and employed the Fe:Mn ratio to reconstruct anoxic conditions in the lake (Nacher et al., 2013).

The chronology was established based on eighteen AMS [14]C measurements (Table 1). Attempts to constrain the chronology of the top core via [210]Pb and [137]Cs gamma assay measurements failed to produce any meaningful results probably due to surface

sediment mixing. An attempt to establish a tephra-based chronology also failed due to a poor match to any known volcanic eruptions. The radiocarbon age estimates were converted into calendar years BP via BACON software (Blaauw and Christen, 2011) using the INTCAL13 data set of Reimer et al. (2013). An age–depth curve was derived based on a smoothing spline model (Fig 2; for details on age depth construction see A1).

### 3.2 Vegetation reconstruction

### 3.2.1 Pollen based quantitative reconstruction of land cover using the REVEALS model

To determine the past vegetation cover we used pollen analysis on samples of 1 cm$^3$ at intervals ranging between 5 and 10 cm (a total of 105 samples) along the composite core. Sediment preparation followed the protocol of Goeury and de Beaulieu (1979). We identified the pollen grains using the atlases of Reille (1995, 1999). A minimum of 300 terrestrial pollen grains were counted at each level and used to calculate the pollen percentages. We corrected for biases in taxon-specific pollen productivities and dispersal, and thereby produced a quantitative reconstruction of the vegetation cover in the region surrounding Lake Oltina using the Regional Estimates of Vegetation Abundance from Large Sites, i.e., REVEALS model (Sugita, 2007). In this model, we used pollen productivity estimates (PPE) for the most characteristic plant taxa from the studied region. For 13 plant taxa that include five woody and eight herbaceous and shrubs taxa we have used PPEs measured in this particular region (Grindean et al., 2019). We have complemented these with literature-based PPEs for 14 additional taxa that significantly contribute to the regional vegetation composition (Table 2). We used the Sugita (2007) dispersal model with default settings for neutral atmospheric conditions and wind speed (3 m/s). The literature-based fall speed of each pollen type (Table 2) is used to model dispersal. The spatial extent of the regional vegetation is set at a 100 km radius. The vegetation cover reconstructed using REVEALS always adds up to 100%, which means that taxa not included in the model, as well as non-pollen producing areas, are ignored. The 27 taxa selected for our REVEALS model represent between 77 and 95 of the percentages in the terrestrial pollen sum. Significant changes in the vegetation assemblages were defined using stratigraphically constrained cluster analyses (incremental sum of squares method) of REVEALS based vegetation cover percentages in Tilia (Grimm, 2004).

### 3.2.2 Leaf wax *n*-alkanes based vegetation reconstruction

To determine the source of organic matter and the predominant vegetation type (Eglinton and Calvin, 1967; Ficken et al., 2000; Diefendorf et al. 2015), we measured the concentration of higher-plant derived *n*-alkane homologues of 60 sediment samples selected along the composite core. *n*-Alkanes are an integral part of higher-plant leaf epicuticular waxes, highly resistant to degradation and among the most stable lipid components of the protective waxes coating terrestrial plant leaves (Eglinton and Eglinton, 2008; Sachse et al., 2012). They are commonly used to distinguish sources of organic matter based on their chain-length (see below). *n*-Alkanes were extracted from freeze-dried and finely ground sediment samples (ca. 1 g dry weight) with a Büchi Speed Extractor at 75 °C and 100 bar using 20 ml of a mixture of dichloromethane/methanol (9:1) for 10 min, which was repeated three times. Total lipid extracts (TLE) were dried under a stream of N2 at 36 °C. The a-polar fraction containing *n*-alkanes was eluted from the TLE by silica-gel column chromatography using hexane. *n*-Alkanes were subsequently isolated from the a-polar fraction using urea-adduction (Vasiliev et al.,2013). *n*-Alkanes homologues were separated and quantified by gas chromatography/mass spectrometry

(GC/MS) using a Thermo Finnigan Trace GC equipped with a HP-5MS column (30 m x 0.25 mm x 0.25 µm) connected to a Thermo Finnigan DSQ II mass spectrometer. The GC oven was held at 70 °C for 1 min, ramped at 10 °C/min to a final temperature of 280 °C, which was held for 15 min. *n*-Alkanes were identified by comparison of their mass spectra and retention time to an external standard (n-C7 to n-C40; Supelco) at a concentration of 25 ng µl. They were quantified using total ion chromatogram peak areas calibrated against the external standard. Precision of the quantification is 96% as inferred from the standard deviation of repeated standard runs (n = 5). Concentrations of individual *n*-alkanes were expressed as ng µl dry weight of sediment.

In this study, we calculated the ratio of straight-chain *n*-alkanes of different chain lengths (homologues) as these have been previously used as proxies for the relative contribution of various types of plants in lacustrine sediments (e.g. Ficken et al., 2000; Zhou et al., 2005). Average chain length (ACL) is an indicator of the relative abundance of short ($C_{16}$-$C_{20}$) vs long chain *n*-alkanes and may be linked to the predominance of higher taxonomic plants over lower taxonomic plants (Ficken et al., 2000; Eglinton and Calvin, 1967). Within the long chain *n*-alkanes, the abundance of *n*-alkanes with *n*-$C_{31}$ and *n*-$C_{33}$ may be indicative of grass predominance, whereas *n*-$C_{27}$ and *n*-$C_{29}$ may indicate a predominantly tree covered landscape (Aichner et al., 2010; Meyers, 2003). The aquatic index ($P_{aq}$) quantifies the abundance of submerged and floating vascular macrophytes, which are characterised by medium chain length *n*-alkanes, relative to emergent plant types that are characterised by long chain *n*-alkanes (Ficken et al., 2000). It should be noted that *n*-alkanes are less successful in detecting coniferous than angiosperms (Diefendorf et al. 2015) and that some overlap within the mid chain length alkanes *n*-$C_{23}$ and *n*-$C_{25}$ is possible (Aichner et al., 2010; Meyers, 2003).

The *n*-Alkanes proxies were calculated using the following equations:

Higher plants ACL C25–C33 =(25 × C25 + 27 × C27 + 29 × C29 + 31 × C31 + 33 × C33)/(C25 + C27 + C29 + C31 + C33), (Poynter et al., 1989)

Tree vs. grass= (C27+C29)/(C31+C33), (Aichner et al., 2010).

$P_{aq}$ = (C23+C25)/(C23+C25+C29+C31), (Ficken et al., 2000).

Analysis of the composition of lipid compounds in modern river sediment deposits along the Danube River shows a predominantly local source of branched glycerol dialkyl glycerol tetraethers derived proxies (Freymond et al., 2017). In line with this finding we presume that changes in *n*-alkane homologue abundance in our record integrate vegetation changes in the lake, near the lake, but also a more regionally in the lake catchment.

### 3.3 Regime disturbances by fire and herbivores

To determine past disturbance by fire, macroscopic charcoal particles were counted on samples of 2 cm$^3$ extracted at 1 cm contiguous intervals. Samples were bleached, wet-sieved through a 150-μm mesh and identified under a stereomicroscope following the methodology described in Feurdean et al. (2017). Here we report only the results for total macro-charcoal particles. We calculated the macro-charcoal accumulation rate (CHAR, particles cm$^{-2}$ yr$^{-1}$) by dividing the total macro-charcoal concentration by sediment deposition time (yr/cm). To determine past grazing activity, coprophilous fungi (*Sporomiella, Sordaria,* and *Podospora*) were tallied during routine pollen counting (van Geel et al., 1980; Bakker et al., 2013). Percentages of coprophilous fungi were determined by adding their own sum to the total terrestrial pollen sum.

### 4. Results

### 4.1 Chronology and sediment composition

The lithology of the core showed little variability throughout the profile and comprises clay, gyttja clay and sandy clay. The age-depth model indicates a rather constant sediment accumulation rate with a mean of 5 yr /cm and no evidence of hiatuses (Fig. 2; A1). The OM as determined from LOI at 550°C varied between 3 and 10% with slightly higher values between 5000 and 3500 cal yr BP and over the past 2000 years (Fig. 3). Our selected geochemical detrital element, Zr, proxy for erosion, showed the lowest values between 5500 and 3500 cal yr BP (up to 150 ppm) and greater, highly fluctuating values at the beginning of the record and over the 2500 years (Fig. 3). The Fe: Mn ratio, proxy for anoxic conditions in the lake, generally displayed large variability throughout the profile, although values were higher between 4000 and 3500 cal yr BP, around 3000 and between 2000 and 1500 cal yr BP (Fig. 3).

### 4.2 Landscape reconstruction from pollen

The pollen record indicated three major periods of change in land cover and vegetation openness over the last 6000 years at Lake Oltina based on the cluster analysis: open temperate deciduous broadleaf forest between 6000 and 4200 cal yr BP; the maximum extent of broadleaf forest tree cover between 4200 and 2500 cal yr BP and the expansion of grassland between 2500 cal yr BP to the present (Fig. 4). Results from the REVEALS model suggest that landscape openness was ca. 10-15% greater than the estimates derived from the raw pollen data. Overall, REVEALS indicate a greater proportion of *Carpinus orientalis*, *Tilia*, *Acer*, Rosaceae, Cerealia and Asteraceae, and a lower proportion of *Corylus avellana, Betula, Ulmus, Alnus*, *Fraxinus, Salix, Artemisia,* Chenopodiaceae than the raw pollen data (Fig. 4). *Quercus, Plantago lanceolata* and Poaceae show a largely similar abundance in both the raw data and REVEALS estimates.

*6000-4200 cal yr BP: Open temperate deciduous broadleaf forest or forest-steppe*

The REVEALS estimate of tree cover fluctuated around 40%, compared to ~ 55% in the raw pollen percentages, and was primarily represented by *Carpinus orientalis, Quercus, Carpinus betulus, Corylus avellana, Tilia* and *Ulmus* (Fig. 4). The REVEALS model predicts an almost equal proportion of total forb (*Artemisia,* Chenopodiaceae, Asteraceae, Rosaceae,

Brassicaceae, *Plantago major*, *Thalictrum,* Caryophylaceae) and grass (Poaceae) in the herbaceous cover (Fig. 4). However, in the raw pollen percentages, forbs dominate the herbaceous assemblages (30%) whereas Poaceae constitutes ~ 20% (Fig. 4; A2). The Cerealia cover estimate fluctuates around ~ 20% in the REVEALS model and was below 5% in the raw pollen percentages. CHAR values were high between 6000 and 5000 cal yr BP and declined markedly thereafter (Fig. 3). The abundance of coprophilous fungi (*Podospora, Sordaria* and *Sporomiella*), on the other hand, rose between 5000 and 4000 cal yr BP (Fig. 3; A2).

*4200-2500 cal yr BP: Maximum extent of temperate deciduous broadleaf tree cover*

Tree cover increased to its maximum extent in the profile (fluctuating around 55%) and was mostly represented by *Carpinus orientalis* and *Quercus* with some occurrence of *Carpinus betulus, Corylus avellana* and *Tilia* (Fig. 4). The rise in *Carpinus orientalis* abundance is more evident in the REVEALS reconstruction (40%) than in the raw pollen percentages (20%). In the herbaceous cover, Poaceae declined the most but there were no marked changes in pollen of primary anthropogenic indicators (Cerealia; Fig. 4). Both, the abundance of CHAR and of coprophilous fungi decreased to one of the lowest in the profile (Fig. 3).

*2500-0 cal yr BP: Decline in Carpinus orientalis-Quercus broadleaf cover; the expansion of grassland/pasture and arable cover*

The tree cover declined abruptly to ca. 20%; this was most evident for *Carpinus orientalis,* from 40% to 10% (Fig. 4). However, tree cover fluctuated strongly over the last 2500 years; with intervals of lower values (20%) between 2500 and 1700 cal yr BP, and over the last 1000 years, and of increases (30%) between 1700 and 1000 cal yr BP (Fig. 4). The REVEALS estimate also suggests an increased proportion of grass (Poaceae), cultivated cereals (Cerealia and *Secale cereale*) and fobs. Among forbs, ruderal taxa Asteraceae, *Plantago lanceolata* and Rosaceae showed the most visible increase (Fig. 4; A2)*.* CHAR values increased gradually from the beginning of this time interval and attained a maximum in the profile between 2000 and 1700 cal yr BP, followed by the lowest values in the profile over the last 1000 years (Fig. 3). The abundance of coprophilous fungi was particularly elevated between 2500 and 2000 cal yr BP and over the last 1000 years.

**4.3 *n*-Alkane based lake catchment ecosystem changes**

The $(C_{27}-C_{29})/(C_{31}-C_{33})$ ratio in Lake Oltina varied between 0.63 and 5.99 and showed greatest values (2.84) between 4200 and 2000 cal yr BP and the lowest values between 5500-4200 cal yr BP (1.76) as well as over the past 2000 years (1.83; Fig. 3). The ACL varied between 27 and 30 and shows an opposite pattern to the $(C_{27}-C_{29})/(C_{31}-C_{33})$ ratio. $P_{aq}$ varied between 2 and 12, and showed the greatest values between 6000-5500 cal yr BP, 4000-2500 cal yr BP and 1200-500 cal yr BP.

**5 Discussion**

**5.1 Forest steppe/woodland between 6000 and 2500 cal yr BP with a maximum tree cover between 4200 and 2500 cal yr BP.**

The pollen-based quantitative land cover reconstruction shows an average tree cover of 40% between 6000 and 4200 cal yr BP and a tree cover maximum of 50% between 4200 and 2500 cal yr BP in the surroundings of Lake Oltina (Figs. 4, 5). In a pollen-based biome reconstruction, such a proportion of trees is likely to be indicative of a forest-steppe or open woodland type (Marinova et al., 2018). This woodland consists of tree taxa of xerothermic character including *Quercus* (likely *Q. cerris*, *Q. pubescens*) and *Carpinus orientalis,* along with temperate trees such as *Carpinus betulus*, *Acer*, *Tilia*, *Ulmus*, *Fraxinus*. The shrubs (Rosaceae, *Rosa canina*, *Sambucus*, *Prunus*, *Cornus*), grasses and forbs communities were abundant and composed of a diverse mixture of mesophytic, xerothermic and halophilous taxa (Fig. A2). Coeval with the maximum extent in tree cover, the *n*-alkanes were dominated by the shorter chain lengths (ALC) and a higher $n(C_{27}+C_{29})/(C_{33}/C_{31})$ ratio, indicative of an increased contribution of tree-derived *n*-alkanes (Meyers, 2003; Aichner et al., 2010). However, the concentration of individual *n*-alkanes $>C_{27}$ (not shown) varied with that of detrital element Zr. It was high between 6000 and 4200 cal yr BP and declined between 4200 and 2500 cal yr BP, suggesting a reduction in the terrestrial plant delivery into the lake during the highest tree cover. The maximum extent of tree cover parallels a slight Fe:Mn ratio increase more evidently between 4000-3500 cal yr BP and 3000-2500 cal yr BP. This may indicate the establishment of more anoxic conditions (Naeher et al., 2013), possibly associated with a higher lake level due to the intensification of Danube water and sediment discharge, a higher lake trophic status, or less turbulent conditions. Increased anoxia linked to a higher lake trophic status and the decomposition of organic matter is supported by the slight increase of submerged aquatic macrophyte ($P_{aq;}$ Fig. 4). The presence of more aquatic plants in the lake at the time of the rise in Fe:Mn ratio is also suggested by the increased abundance of aquatic/wetland taxa (*Potamogeton*, *Myriophilum*, *Typa /Sparganim*). On the other hand, low values of the lithogenic element Zr between 4200 and 2500 cal yr BP indicate more stable slope conditions with low run off, which might support the hypothesis that increased anoxia may have resulted from less turbulent conditions in the lake.

On a regional scale, the 6000-4200 cal yr BP interval was characterised by contrasting climate conditions north and south of 45 degrees latitude, due to differences in the dynamics of the storm tracks carrying moisture from the North Atlantic Ocean and Mediterranean Sea (Persoiu et al., 2017). Whilst southern Europe lake levels were low prior to 4500 cal yr BP, those in central Europe showed high stands during the same period (Magny et al., 2013). An opposite pattern is, however, visible after 4500 cal yr BP, when southern Europe lake levels increased whereas those from central Europe declined. The lake level increase after 4500 cal yr BP, also paralleled an intensification of fluvial activity in several rivers in southern Romania and southern part of the Danube river at this time (Bozilova and Tonkov, 1998; Filipova-Marinova et al., 2007; 2011; Howard et al., 2004; Persoiu, 2010). Coastal lakes along the Black Sea were also at higher water levels at this time, whereas the Black Sea's level fluctuated strongly (Lamy et al., 2006). Lake Oltina is situated at 45º N and thus at the transition between these contrasting southern and northern Europe changes in hydro-climatic conditions. Our evidence of maximum expansion of the tree cover, and possibly an association with wetter conditions after 4500 cal yr BP may indicate the response of tree cover to

increased moisture availability in the region (Fig. 3). A greater-than-present tree cover at 6000 cal yr BP in response to higher precipitation has also been simulated for this region (Patriche et al., 2020).

Macrocharcoal based reconstruction of biomass burning, and thus disturbance by fire, was low between 4200 and 2500 cal yr BP at the time of greater forest cover and wetter climate conditions (Fig. 3; Diaconu et al., 2018). This fire–climate-vegetation relationship is typical for a temperate environment, but contrasts with the pattern found in environments with low vegetation

productivity, where increased moisture tends to enhance vegetation productivity and therefore fuel availability (Pausas and Ribeiro 2013; Feurdean et al., 2020). Disturbance by herbivores, as inferred from the abundance of coprophilous spores, showed moderate values at the beginning of the record, but declined during the interval of tree cover increase (Fig. 3). This may point to some impact by herbivores on the degree of forest openness, i.e, increased tree cover with decline in grazing activity. However, given the very large size of the study lake, the long distance from the lakeshore to the coring 1point might

have limited transportation of these spores and impact on how representative their presence might be (Bakker et al., 2013).

**5.2 Transition from forest steppe to cultural steppe over the last 2500 cal yr BP**

Tree cover declined from 50% at 2500 cal yr BP to ~ 20% at 2200 cal yr BP (Fig. 5). Tree species composition retained a xerothermic character, although *Carpinus orientalis*, typical of hot and dry air and soil conditions (Sikkema and Caudullo,

2017), declined most strongly from 40% to 10% (Fig. 4). The pollen-based decline in tree cover was concurrent with decreasing $n(C_{27}+C_{29})/(C_{33}/C_{31})$ ratio, characteristic of an increased contribution of grasses, further suggesting a tree loss (Fig. 4). An average of 20% tree cover at 2200 cal yr BP is close to the current tree cover in the surroundings of Lake Oltina (15%), which suggests the opening up of the forest steppe to a similar extent as today. Anthropogenic conversion of broadleaf forests to agricultural land from 2500 cal yr BP onwards is suggested by the rise in the pollen of Cerealia (*Secale cereale*, *Triticum*, *Zea*,

*Hordeum*), grassland (Poaceae), pastoral and ruderal indicators (*P. lanceolata, Urtica, Rumex,* Chenopodiaceae, *Artemisia,* Asteraceae, Apiaceae; Figs. 4, 5; A2). The levels of pollen of grazing indicators and nutrient-enriched soils (*Plantago lanceolata, Urtica, Rumex),* as well as coprophilous fungi, are particularly elevated after 2200 cal yr BP and may reflect more intensive animal husbandry. A higher representation of Cerealia in the REVEALS estimate (40%) than the raw pollen record (5%) is not surprising, given the poor productivity and dispersal of most of Cerealia pollen types. However, the magnitude of

this difference is greater than generally reported in the literature. Nevertheless, the good match between the proportion of arable cover in the REVEALS and current land cover maps from CORINE for recent times suggest that our REVEALS reconstruction best reflects that of cultivated land. Uncertainties in pollen-based land cover reconstruction are common and connected to the availability and accuracy of the productivity estimates (PPE), changes in cropland and grassland management and the general assumptions of the REVEALS model (Sugita et al., 2017; Hellman et al., 2009; Trondman et al., 2015;

Feurdean et a., 2017b). Our Cerealia pollen includes *Triticum, Zea, Hordeum* and *Secale cereale*, for which we have used productivity estimates derived from the calibration of local surface pollen samples with a vegetation inventory (Grindean et al., 2019). These productivity estimates are considerably lower than the average for Europe (0.22 vs. 1.85; Mazier et al., 2012) and therefore the main cause of the high proportion of Cerealia cover reconstructed by REVEALS and for the disparity between

the outcome of this study and elsewhere in Europe. Furthermore, the crop species included in Cerealia also vary regionally and with time, which may also introduce variation when applying their PPEs for landscape reconstruction over an extended period of time. Lastly, the occurrence of wild grass species with pollen that fall in the Cerealia pollen type (all Poaceae grains larger than 40 μm), may have led to an overestimation of the proportion of Cerealia at certain times in the past. Biomass burning increased concurrently with the spread of pasture/grassland communities, which may indicate the use of fire for land use management (Fig. 3). However, fire activity fell to its lowest level in the profile in the last 1000 years, probably due to a decline in biomass availability associated with intensive land use and landscape fragmentation typical of lowlands (Marlon et al., 2016; Feurdean et al., 2013; 2020).

The timing of forest loss coincides with the peak number of archaeological finds in the Iron Age but is pre-dated by the abundant archaeological finds of the Bronze Age i.e., 4000 cal yr BP (Fig. 5). Interestingly, the typology of the houses in this area i.e., small houses three-quarters buried in the ground with one-quarter above ground comprising mud bricks walls, roofed with straw or reeds typical during the past 3000 years also reflects the limited availability of timber for building (Ailincăi, 2009). Historical records show that, due to its smaller size, *Carpinus orientalis* was managed in coppice stands for household items, fuel production as firewood or charcoal (Goldsteinn et al., 1990; Atlas of European Tree Species, 2017) and this may have also been the cause of its marked reduction from 2500 cal yr BP at Lake Oltina. *Quercus* decline on the other hand, was more modest except for the last 1000 years. Historically, the aftermaths of the Slavic (first millennium AD) and Mongol (1224 AD) invasions could be a reason for a strong *Quercus* decline around this time (Matei, 1984; Epure, 2004). Oak was preferentially used for fortification in SW Romania (Krause et al., 2019; Gumnior et al., 2020). Nevertheless, tree cover, and *Quercus* in particular, recorded episodic rises with the strongest being visible between 1700 and 1300 cal yr BP. This reforestation phase appears to coincide with the post-Roman decline in rural settlement and the subsequent re-growth of secondary forests (Roberts et al., 2018). However, selective oak preservation, due to its economic significance, may have also led to this pattern of *Quercus* increases, as documented over large areas in Europe (Garner et al., 2002; Feurdean et al., 2011; Jamrichová et al., 2017; Gumnior et al., 2020). Models of deforestation rates, using a scenario that accounts for population history and technological advances, suggest that the extent of deforestation in the lower Danube basin increased continuously from 4000 cal yr BP and rapidly doubled after 1000 AD (Kaplan et al., 2009; Giosan et al., 2012), thus much later than our pollen-based land cover reconstruction. Assuming no age correction for the hard water effect would result in an age 1000 years older than that calculated by taking into account this dating limitation, which pushes the timing of major deforestation back to 3500 cal yr BP. This corroborates with the timing of deforestation found in Transylvania, central Romania based on REVEALS model (Feurdean et al., 2015) and with the onset of local increase in archaeological finds of the Bronze Age (Fig. 5).

In-lake and catchment changes are also apparent in the Lake Oltina around the onset of forest loss. A slight increase in the Fe:Mn ratio, along with the sharp rise in Zr concentrations between 2500 and 1500 cal yr BP may reflect soil erosion and detrital input into the lake associated with a diminished tree cover (Fig. 3), an inference also supported by an abrupt rise in charcoal and *n*-alkanes $> C_{27}$ concentrations. Intensification in fluvial activity in several rivers in southern Romania (Howard et al., 2004; Persoiu and Radoane, 2017), as well as on a wider scale in Central Europe (Wirth et al., 2013), has been

reconstructed after 3000 cal yr BP. Danube's water and sediment discharged could therefore be another source of detrital and terrestrial plant delivery into the lake. In contrast, the Black Sea level lowered, which also suggests drier conditions in Eastern

Mediterranean and over the Black Sea around the time of deforestation (Lamy et al., 2006), in line with the drier conditions in SW Romania found between 2000 and 1000 cal yr BP (Dragusin et al., 2014). Southern European lake levels remained high at the time of deforestation, while those from central Europe (Magny et al., 2011; 2013) and central Romania declined (Feurdean et al., 2013), though others in Romania increased (Magyari et al., 2009).

**5.3 Comparison with other European forest steppe regions**

Our quantitative record of vegetation cover indicates a higher than present tree cover across the landscape of the eastern Lower Danube Plain between 6000 and 4200 cal yr BP with an absolute maximum of 50% (60% raw pollen percentages) between 4200 and 2500 cal yr BP (Fig. 6). The composition and structure of mid to late Holocene vegetation of the eastern Lower Danube Plain resembled, to a large degree, that of other European forest steppe areas, although particularities also exist. Whilst

in central eastern European forest steppe, *Quercus* and *Carpinus betulus* were the dominant tree species with a lower occurrence of *Tilia, Ulmus, Corylus* and *Pinus* (Magyari et al., 2010; Feurdean et al., 2015; Kuneš et al., 2015), on the Eastern European Plain, *Tilia* and *Quercus*, and in some places *Pinus,* were the dominant tree taxa (Kremenetski et al., 1995; Novenko et al., 2016, 2018; Shumilovskikh et al., 2018, 2019). Forests in the Black Sea region also included thermophilus taxa i.e., *Quercus cerris*, *C. orientialis* (this study, Marinova et al., 2006; Tonkov et al., 2014) Remarkably, *Carpinus orientalis* was

found to be significantly more abundant around the Black Sea coast i.e., Romania (20%), Bulgaria (>5%), or with only scattered occurrences in Ukraine, whilst it was absent from Central European forest steppe. Our pollen-vegetation calibration model shows that the adjusted abundance of *C. orientalis* is twice that recorded by its raw pollen percentages (Fig. 4).

Mid-Holocene landscape openness near Lake Oltina (ca. 45% raw pollen percentages) appears to fall in between that of the steppe in south-eastern Bulgaria (60-80%; Tonkov et al., 2014), the forest steppe of Ukraine (25%; Kremenetski et al., 1995;

1999) and that of the Eastern European Plain (20-50%; Shumilovskikh et al., 2018). However, landscape openness was greater than in other forest steppe sites from central eastern Europe, i.e., Romania (Feurdean et al., 2015; Tantau et al., 2006), Hungary (Willis et al., 1997; Willis, 2007; Magyari et al., 2010), Czechia and Slovakia (Pokorny et al., 2011, 2015; Hajkova et al., 2013; Kuneš et al., 2015) where it varied between 10 and 35 % (Fig. 6). The composition of herbaceous plant cover included grasses (Poaceae) and a diversity of forbs thriving on a wide variety of habitats ranging from dry and saline soils (*Artemisia,*

Chenopodiaceae, Asteraceae Compositae, Asteraceae Tubuliflorae) to dry and wet meadows (*Filipendula, Galium, Anthemis, Aster*, Caryophyllaceae, *Euphorbia, Helianthemum, Hypericum*, Fabaceae, *Plantago lanceolata, P. major /P. media, Teucrium, Thalictrum* and *Verbascum;* Fig. 4; A2). Notably, however, the proportion of steppe and saline elements (*Artemisia,* Chenopodiaceae) was greater at sites located in the Black Sea region, in agreement with greater temperature seasonality, lower precipitation as well as the occurrence of saline soils in this region.

The comparison of pollen records from the European forest steppe shows a west to east gradient in the timing and magnitude of deforestation (Fig. 6). For example, the timing of major anthropogenic ecosystem transformation in the Lower Danube Plain

from about 2500 cal yr BP fall in between that of other records in lowland areas in central eastern and south eastern Europe, where it generally occurred after 3000 cal yr BP (Fig. 6). However, this is earlier than on the Eastern European Plain Europe, where it mostly occurred after 2000 cal yr BP. On the Thracian Plain, south eastern Bulgaria, anthropogenic deforestation was,

however, noted already from 4000 cal yr BP (Connor et al., 2013). The anthropogenically-driven opening up of the forest steppe soon reached a similar extent as today in most regions, which then remained open until the present day, although the climate conditions could have allowed the recovery of tree cover. Notably, however, the study region is increasingly confronted by desertification (European Environmental Agency, 2016). Given its dry character, the conversion of forests to cropland may have acted as a positive feedback to the warm and dry climate, enhancing evaporation, altering the moisture balance, further

contributing to the tendency towards the aridisation of Lower Danube landscapes. Ongoing climate change (warmer temperatures and a decline in precipitation), coupled with agricultural intensification is likely going to exacerbate the process of desertification.

**6 Conclusions**

The pollen-based vegetation modelling applied here (REVEALS) provides the first, long-term quantitative reconstruction of land cover changes across the Lower Danube Plain (SE Romania) and in SE Europe. Enhanced moisture availability likely led to a more extensive tree cover between 6000 and 2500 cal yr BP and its maximum of 50% between 4200 and 2500 cal yr BP. This woodland consists of tree taxa of xerothermic character including *Quercus* (likely *Q. cerris, Q. pubescens*) and *Carpinus orientalis,* along with temperate trees such as *Carpinus betulus*, *Acer, Tilia, Ulmus, Fraxinus*. However, the proportion of

xerophilous tree taxa, *C. orientalis,* increased between 4200 and 2500 cal yr BP.  The forests of the Lower Danube Plain were intensively cleared and converted to agricultural land and pasture/semi-natural grasslands from the Iron Age (2500 cal yr BP). The landscapes become deforested to present-day levels (ca. 20% tree cover) 2200 years ago. Tree cover remained low throughout the last two millennia demonstrating the continuous anthropogenic pressure on the surrounding region. The permanent loss of the tree cover is visible across a west-east gradient of the Central Eastern European forest-steppe region

highlighting its sensitivity to anthropogenic impact. Given the dry character of the study region, deforestation and land conversion to agriculture may have additionally enhanced evaporation altering the moisture balance and further contributing to the tendency towards the aridisation of Lower Danube landscapes. This palaeoecological study also demonstrates that, at broad spatial scale, the natural vegetation of the eastern Romanian Plain under climatic conditions similar to today is forest steppe/woodlands, which is in agreement with expert-based assessment of potential tree cover. However, tree cover extent and

composition have neither been stable in time, nor solely shaped by the climate with disturbances by fire and grazing, and later by anthropogenic impact, playing an important role. In comparison to pollen-based vegetation reconstruction, the PNV assumes a lower proportion of *C. orientalis,* and higher proportion of *Quercus.*

We also show that both the extension and decline in tree cover determined by pollen, a well-established proxy for past vegetation change, is also reflected in the *n*-alkanes record, which indicates their potential as a reliable record of tree versus grass cover changes in dry regions where reliable pollen records are difficult to obtain.

## Data sets

### Accessibility Statement

All essential input data will be made open-access and available online in suitable repositories (Pangaea) upon publication.

**Author contribution**: AF designed the study. AF, AD, AP and MB performed the fieldwork; RG, IT, AF pollen analysis; GF and EMN biomarkers; GF and SMH geochemistry; AD macrocharcoal; ST, AP and AF C14 measurements and age-depth model. RG, AN, AF REVEALS modeling. AF prepared the manuscript with contributions and input from all authors.


**Competing interests**: The authors declare that they have no conflict of interest.

**Acknowledgements:** We thank the managers at Lake Oltina and NATURA 2000 for granting the access and facilitating access to the lake for sediment sampling. G. Florescu and E. Niedermeyer thank U. Treffert for laboratory support of biomarker
analysis, and R. Grindean thanks S. Farcas for granting laboratory access for pollen preparation. We thank Rebecca Kearney for her work on tephrostratigraphy, Mihaly Molnar for the suggestions on the construction of the age the model and David for assistance in field. Finally, we thank the two reviewers Natalie Schroeter and Simon Connor for their constructive comments on the manuscript.


**Financial support**. This work was supported from the Deutche Forschungsgemeinschaft (FE_1096/4) and CNCS-UEFISCDI (PN-II-RU-TE-2014-4-2445 and PN-III-P4-ID-PCE-2016-0711). Support from grant FE_1096/6 during the writing stage is also acknowledged.

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

805

**Figure legends and embedded figures**

**Figure 1.** Potential natural vegetation cover in Europe showing the extent of the European forest steppe region (after Bohn 2003; ©BfN, Bundesamt fur Naturschutz) the location of the study site and the other published sites used for comparisons. 1. Lake Oltina (study site); 2. Lake Vracrov; 3. Sarló-hát; 4. Lake Știucii; 5. Durankulak-2; 6. Durankulak-3; 7. Dovjok; 8. Kardashinski; 9. Sudzha; 10. Selikhovo; 11. Isotechek; 12. Podkosmovo. For references see Table 3.

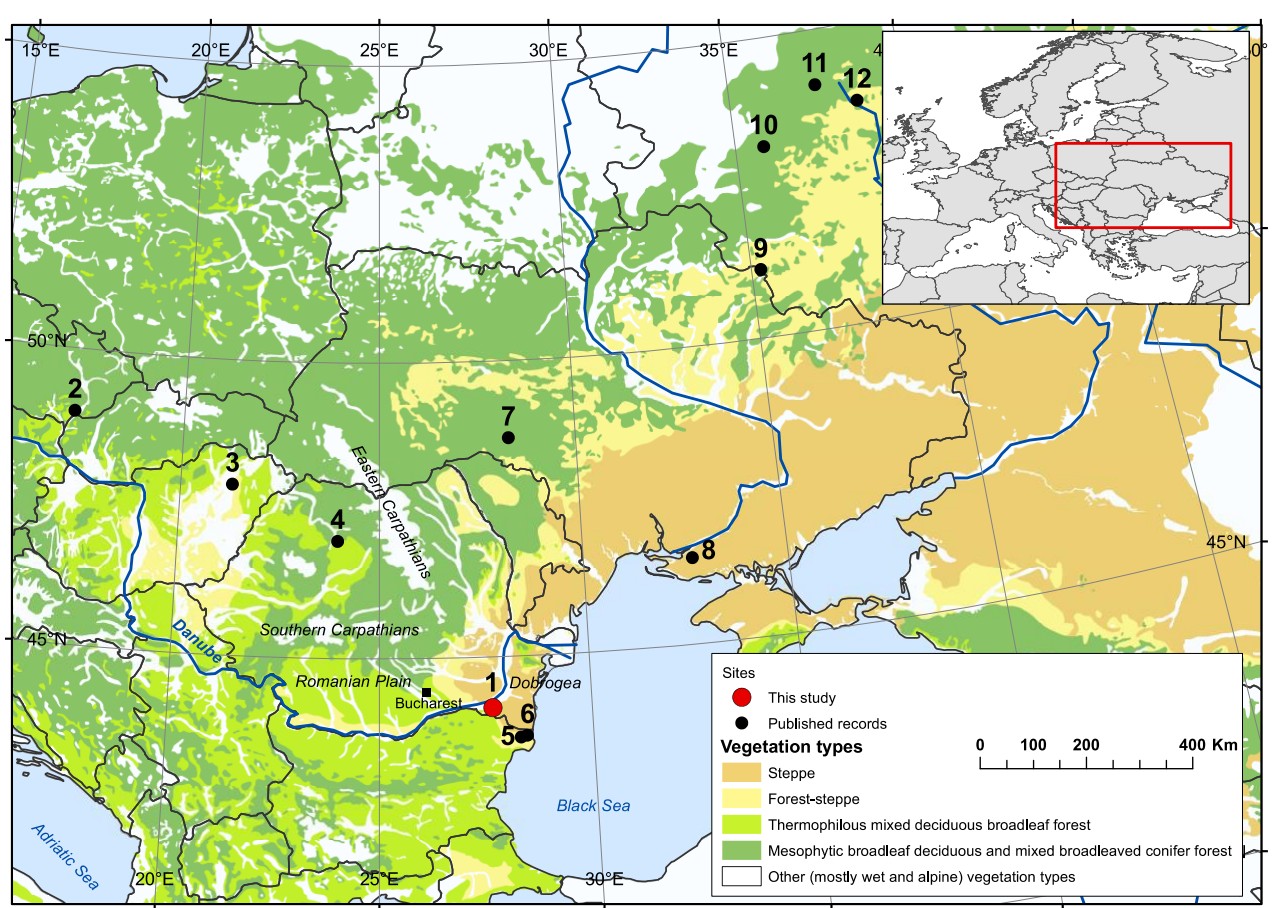

815

**Figure 2**. Age depth model for Lake Oltina using a Bayesian approach (for more see S2).

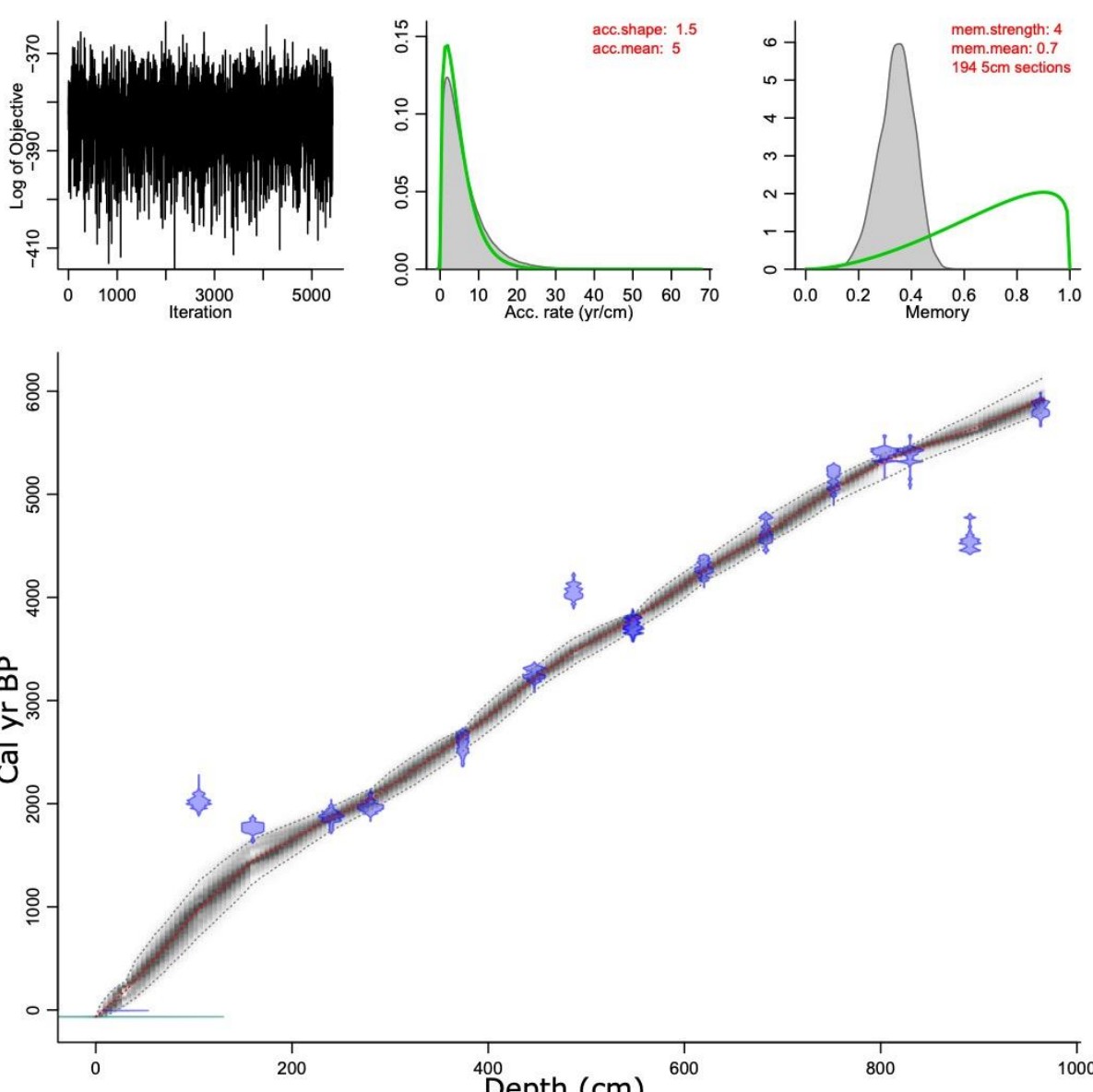

825

**Figure 3**. Integrative diagram showing in-lake ecosystems and catchment changes. Lake properties: volume magnetic susceptibly, organic matter content, and detrital elements (Zr and Fe/Mn ratio). Biomass burning as determined from the macro-charcoal (CHAR) record and grazing activity from coprophilous spores. Landscape cover determined from the arboreal pollen (AP) and the open land pollen percentages; ACL (Average Chain Length) of higher plant and the ratio of ($n$-$C_{31}$ + $n$-$C_{33}$)/($n$-$C_{27}$ + $n$-$C_{29}$) as a proxy for abundance of higher plants, and herbs versus trees predominance in the landscape, respectively. Regional lake level fluctuations determined from the CA score at Lake Ledro, N Italy 45º N and Lake Preola, S Italy 37º N (Magni et al., 2013). Archaeological finds include settlements and cemeteries within a 20-km radius of Lake Oltina taken from Repertoriul Arheologic Național [Archaeological Repertorium of Romania]: http://ran.cimec.ro.

835

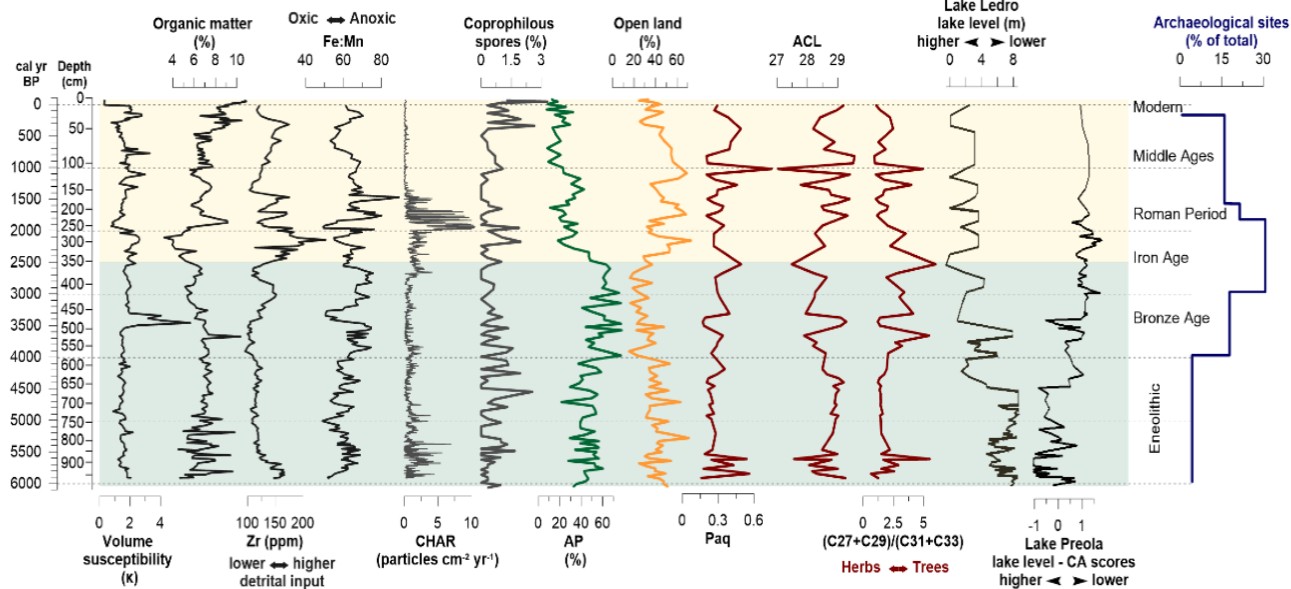

840

**Figure 4.** Raw pollen percentages (A) and estimated regional vegetation cover based on the REVEALS model (B) for 27 taxa including trees, shrubs and herbs at Lake Oltina. Vertical lines denote the timing of the most important changes in the vegetation assemblages.

# Lake Oltina
raw pollen percentages

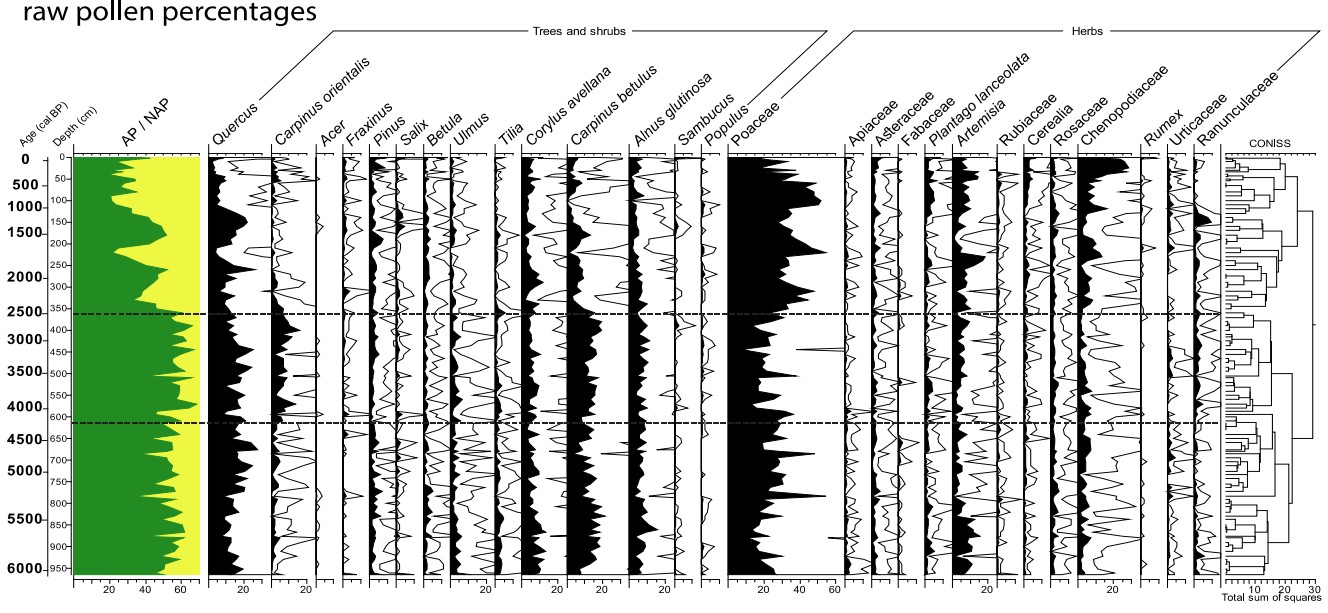

## REVEALS

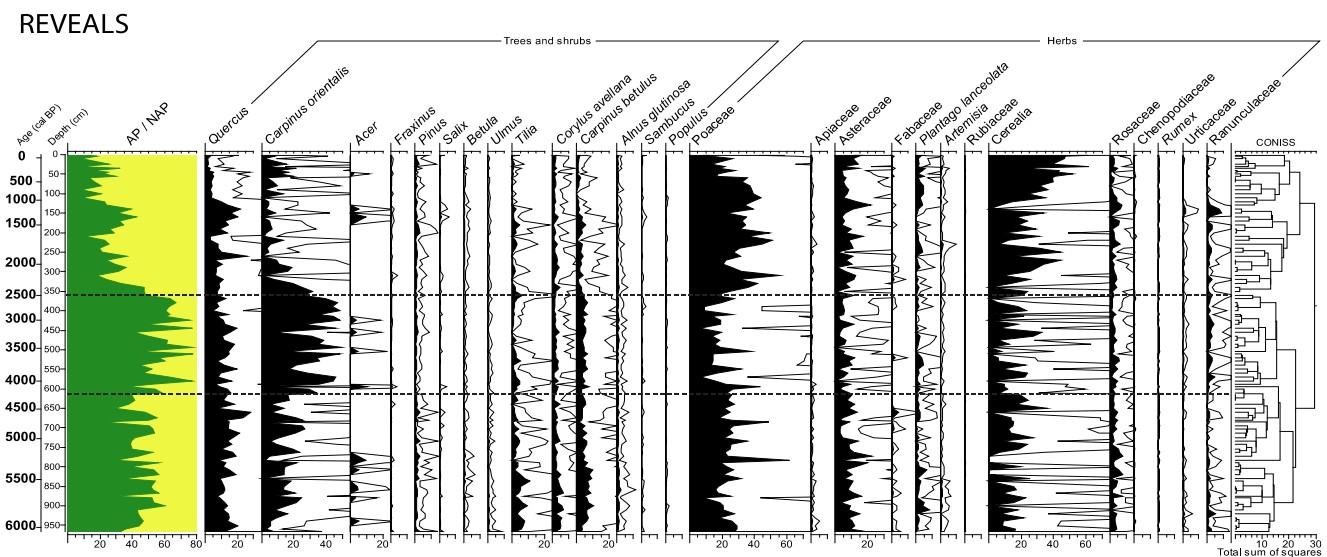

**Figure 5**. Comparative summary of the percentage vegetation cover estimates based on the REVEALS model and raw pollen percentages at Lake Oltina. Open land cover includes all non-arboreal pollen types, mostly indicators of pastures and grasslands. The Cerealia group include *Secale cereale*, *Triticum, Zea* and *Hordeum*. Vertical lines denote the timing of the most important changes in the vegetation assemblages. Archaeological finds as in Figure 3.

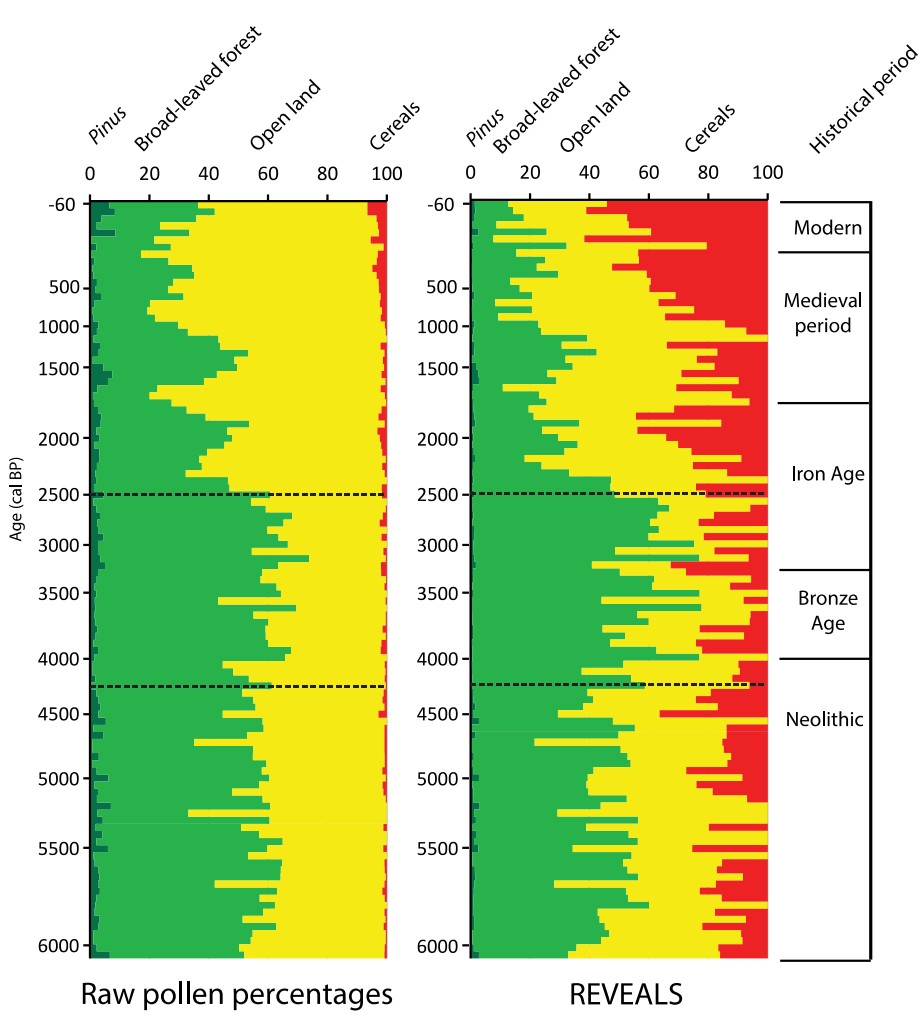

**Figure 6**. Arboreal pollen percentages illustrating temporal trends in deforestation in three different sub-regions along a west east transect across the European forest steppe region. For the location of individual sites see Figure 1 and Table 3.

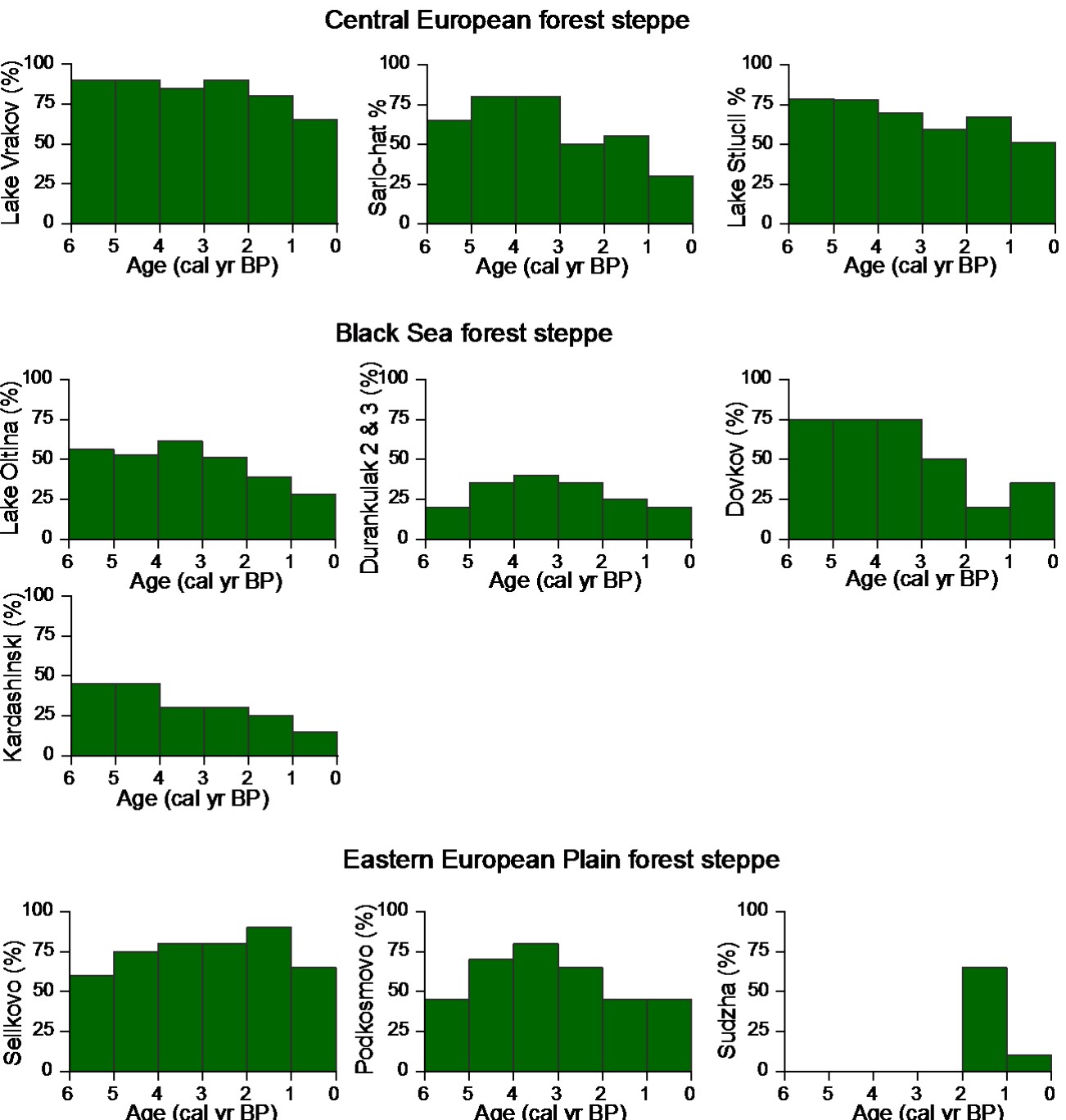

**Tables**

Table 1. AMS $^{14}$C measurements at Lake Oltina.

| Lab. no. | Core | Depth (cm) | Mat. datedC$^{14}$ | $^{14}$C age (±1 sigma) |
|---|---|---|---|---|
| DeA-10928 | 1.1 | 30 | Shell | 880±19 |
| DeA-11083 | 1.2 | 105 | Bulk | 2053±28 |
| DeA-11085 | 1.2 | 160 | Bulk | 1832±27 |
| DeA-11087 | 1.3 | 240 | Bulk | 1928±37 |
| RoAMS 131.45 | 1.3 | 280 | Shell | 3016±30 |
| RoAMS 128.45 | 1.4 | 374 | Shell | 3473±23 |
| RoAMS 132.45 | 1.5 | 447 | Shell | 4042±24 |
| RoAMS 366.45 | 1.5 | 487 | Shell | 4715±34 |
| RoAMS 133.45 | 1.6 | 548 | Plant macro | 3459±31 |
| RoAMS 356.45 | 1.6 | 547 | Shell | 4457±35 |
| RoAMS 134.45 | 1.7 | 620 | Shell | 4856±25 |
| RoAMS 364.45 | 1.7 | 683 | Shell | 5097±36 |
| RoAMS 129.45 | 1.8 | 752 | Shell | 5476±25 |
| RoAMS 135.45 | 1.9 | 804 | Shell | 5648±27 |
| RoAMS 130.45 | 1.10 | 891 | Shell | 5055±30 |
| RoAMS 363.45 | 1.10 | 926 | Bulk | 8886±41 |
| RoAMS 361.45 | 1.10 | 963 | Shell | 6093±42 |

Table 2. Pollen productivity estimates relative to Poaceae and their respective fall speeds used in the REVEALS model. *standardized after Mazier et al. (2012).

a) Local pollen productivity estimates (PPE)

| Taxon | PPE (Grindean et al., 2019) | Fall speed (m s$^{-1}$) | Reference for fall speed |
|---|---|---|---|
| *Quercus* | 1.10 | 0.035 | Mazier et al. (2012) |
| *Carpinus orientalis* | 0.24 | 0.042 | Mazier et al. (2012) |
| *Acer* | 0.30 | 0.056 | Mazier et al. (2012) |
| *Fraxinus* | 2.99 | 0.022 | Mazier et al. (2012) |
| Poaceae | 1.00 | 0.035 | Mazier et al. (2012) |
| Apiaceae | 5.91 | 0.042 | Mazier et al. (2012) |
| Asteraceae | 0.16 | 0.029 | Mazier et al. (2012) |
| Fabaceae | 0.40 | 0.021 | Commerford et al. (2013) |
| *Plantago lanceolata* | 0.58 | 0.029 | Mazier et al. (2012) |
| *Artemisia* | 5.89 | 0.014 | Abraham and Kozakova (2012) |
| Rubiaceae | 7.97 | 0.019 | Broström et al. (2004) |
| Cerealia | 0.22 | 0.06 | Mazier et al. (2012) |
| Rosaceae | 0.29 | 0.018 | Mazier et al. (2012) |

b) Literature-based pollen productivity estimates

| Taxon | PPE | Fall speed (m s$^{-1}$) | Reference |
|---|---|---|---|
| *Pinus* | 2 | 0.031 | Abraham et al. (2014) |

| | | | |
|---|---|---|---|
| *Salix* | 2.31 | 0.022 | Abraham et al. (2014) |
| *Betula* | 2.62 | 0.024 | Abraham et al. (2014) |
| *Ulmus* | 6 | 0.032 | Abraham et al. (2014) |
| *Tilia* | 0.5 | 0.032 | Abraham et al. (2014) |
| *Corylus avellana* | 1.4 | 0.025 | Abraham et al. (2014) |
| *Carpinus betulus* | 3.55 | 0.042 | Abraham et al. (2014) |
| *Alnus glutinosa* | 4.2 | 0.021 | Abraham et al. (2014) |
| *Sambucus* | 1.3 | 0.013 | Abraham and Kozakova (2012) |
| *Populus* | 3 | 0.025 | Matthias et al. (2012) |
| Chenopodiaceae | 4.28 | 0.019 | Abraham and Kozakova (2012) |
| *Rumex* | 2.14 | 0.018 | Mazier et al. (2012) |
| Urticaceae | 10.52 | 0.007 | Abraham and Kozakova (2012) |
| Ranunculaceae | 1.96 | 0.014 | Mazier et al. (2012) |

**Table 3**. Compilation of palaeoecological records along a west east transect of the European Forest steppe. Key: Cz=Czechia; Hu-Hungary; RO-Romania; BG-Bulgaria; UK-Ukraine; RU-Russia.

| Site | Country | Lat/Long | Elevation (m asl) | References |
|---|---|---|---|---|
| 1. Lake Oltina | RO | 44°09'N  27°38'E | 7 | This study |
| 2. Lake Vracrov | CZ | 48°58'N  17°11'E | 100 | Kunes et al., 2015 |
| 3. Sarló-hát | Hu | 47°54'N  21°18'E | 86 | Magyari et al., 2010 |
| 4.Lake Stiucii | RO | 46°58'N  23°58'E | 239 | Feurdean et al., 2015 |
| 5. Durankulak 2 | BG | 43°39'N  28°31'E | <10 | Bozilova & Tonkov ,1995 |
| 6. Durankulak 3 | BG | 43°39'N  28°31'E | <10 | Marinova, 2006 |
| 7. Dovjok | UK | 48°45'N  28°15'E | <100 | Kremenetski ,1995 |
| 8. Kardashinski | UK | 46°31'N, 32°37'E | <100 | Kremenetski ,1995 |
| 9. Sudzha | RU | 51°08' N 35°46' E | 134 | Shumiloviskhk et al., 2018 |
| 10.Selikhovo | RU | 53°13' N 35°46'E | 200 | Novenko et al., 2016 |
| 11.Isotechek | RU | 54°41 N 37°30 E | 200 | Shumiloviskhk et al., 2018 |
| 12.Podkosmovo | RU | 53°40 N 38°35'E | 200 | Novenko et al., 2014 |

**Appendix A1. Lake Oltina chronology**.

A chronology for Lake Oltina was established on the basis of 17 AMS [14]C measurements (Table 1a). Attempts to constrains the chronology of the top core via [210]Pb and [137]Cs gamma assay measurements failed to produce any meaningful results probably due to sediment mixing. An attempt via the geochemical identification of tephra also failed due to a poor match to any know volcanic eruption. The radiocarbon age estimates were converted into calendar years BP via BACON software (Blaauw and Cristen 2011) using the INTCAL13 data set of Reimer et al. (2013). An age–depth curve was derived based on a smoothing spline model. Calendar age point estimates for depths were based on weighted average age-depth curves and also

by taking into account the error range of the calibrated ages. Due to the lack of plant macrofossils, we used mostly shells and bulk material for the radiocarbon measurements. Consequently, age determination was problematic due to the low organic carbon levels and possible hard water effects (Table 1a).

       We have attempted to correct for the reservoir effect in the following ways. Firstly, we compared our youngest radiocarbon date on the shell sample from 30 cm in depth (880 uncal. yr BP) with the potential sediment age of recent samples

based on geochemistry, mineral magnetic measurements and specific pollen marker. From 30 cm geochemical elements (particularly Pb) potentially associated with regional industrialisation (post-1850) show concentrations above the background levels that would naturally be found suggesting an additional anthropogenic input (Fig. A1). Further, mineral magnetic properties (X) also show an increase from 30 cm that might reflect an anthropogenic influence on the sediment (Fig. A1). Rose et al. (2009), Akinyemi et al. (2013) and Hutchinson et al. (2016) note in the Romanian Carpathian Mountains atmospherically

derived inputs of trace elements and heavy metals and mineral magnetic particles from the start of the 20[th] century with peaks from the 1950s. Similarly, Begy et al. (2012) attribute peaks in heavy metals in a lake in the Danube delta to industrial and traffic pollution from the 1950s. We noted the occurrence of pollen of *Ambrosia,* an invasive species that arrived and spread post 1850, also increased at this depth (Fig. A1). Taken together, results from the geochemistry, mineral magnetic measurements and pollen agree in suggesting that the age of the sediment at 30 cm must be post-1850 AD. Based on the

difference of the two age estimates (880 years on the shell) and ~50-100 years (via the geochemistry, mineral magnetic measurement and pollen), the age offset at 30 cm is about 700 years.

       Secondly, for older sediments, we compared the radiocarbon date of the terrestrial macrofossil sample at a depth of 548 cm (3459±3) with the shells from the same layer (4457±35). Here the age difference between the two measurements is 1000 years, which is close to the 700 years age offset observed at 30 cm. Consequently, we have estimated the hard water

effect at about 800 +/- 200 yrs. In the Bacon model an age offset of 1000 years was specified for all radiocarbon dating on shells (Fig. 2). Furthermore, all measurements performed on bulk samples were rejected from the age-depth model. This is because the reservoir effect on bulk sediment is much larger than for shell, with a much larger possible error. In addition, in at least two age measurements on bulk samples (at 150 and 230 cm) the H-fraction was very different from L fraction, which indicates that the bulk organic matter is a composite of very different age organic material. The results of our final model

provide an age-depth curve with fewer age depth reversals than seen when including the bulk samples.

Table 1a. AMS [14]C measurements at Lake Oltina showing different age of the H-and L-fractions of bulk samples for two of the four samples performed using bulk material.


| AMS [14]C Lab Code | HEKAL Sample Nr. | Depth and Mat. Dated | Conventional [14]C age (yr BP ± 1sigma) |
|---|---|---|---|

| | | | |
|---|---|---|---|
| DeA-10928 | I/1451/1 | 20 cm (shell) | $880 \pm 19$ |
| DeA-11083 | I/1451/2L | 95 cm bulk | $2053 \pm 28$ |
| DeA-11084 | I/1451/2H | 95 cm (bulk) | $2818 \pm 42$ |
| DeA-11085 | I/1451/3L | 150 cm (bulk) | $1832 \pm 27$ |
| DeA-11086 | I/1451/3H | 150 cm (bulk) | $2374 \pm 27$ |
| DeA-11087 | I/1451/4L | 230 cm (bulk) | $1928 \pm 37$ |
| DeA-11088 | I/1451/4H | 230 cm(bulk) | $2013 \pm 24$ |

Figure A1. Selected geochemical elements (Pb, Zn), mineral magnetic measurements (magnetic susceptibility) and pollen (*Ambrosia*) displaying their simultaneous increase after 30 cm (vertical dashed line) reflecting the post 1850 AD trend of regional industrialisation and the known spread of an invasive weed.

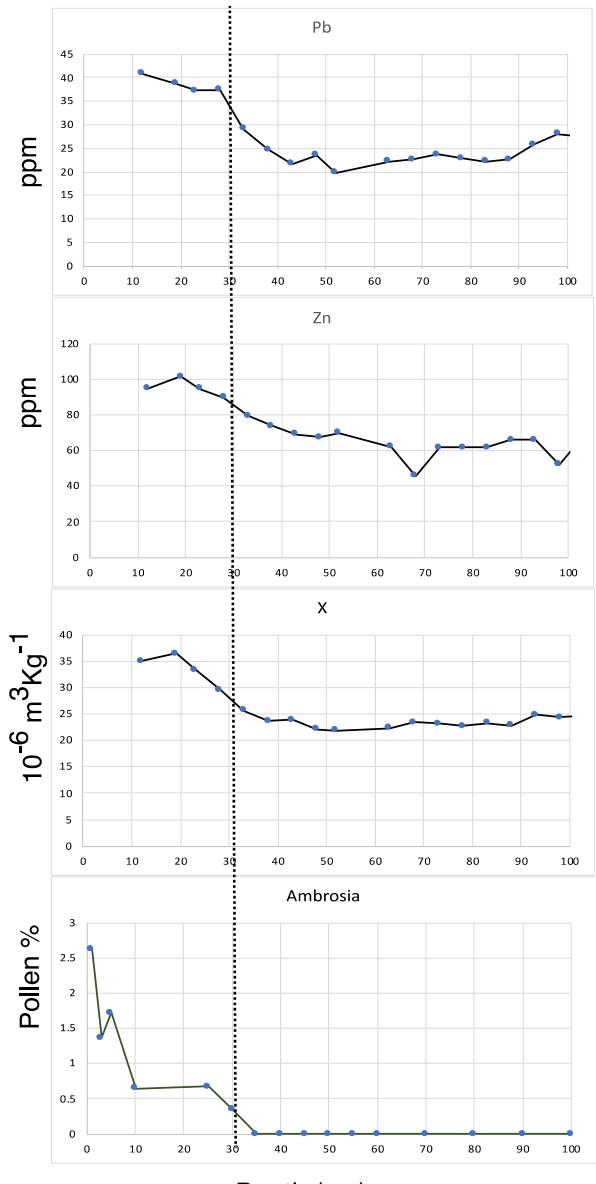

Depth (cm)



**Figure A2.** Full pollen diagram for Lake Oltina grouped on trees, shrubs herbs, wetland and aquatic taxa as well as coprophilous fungi.


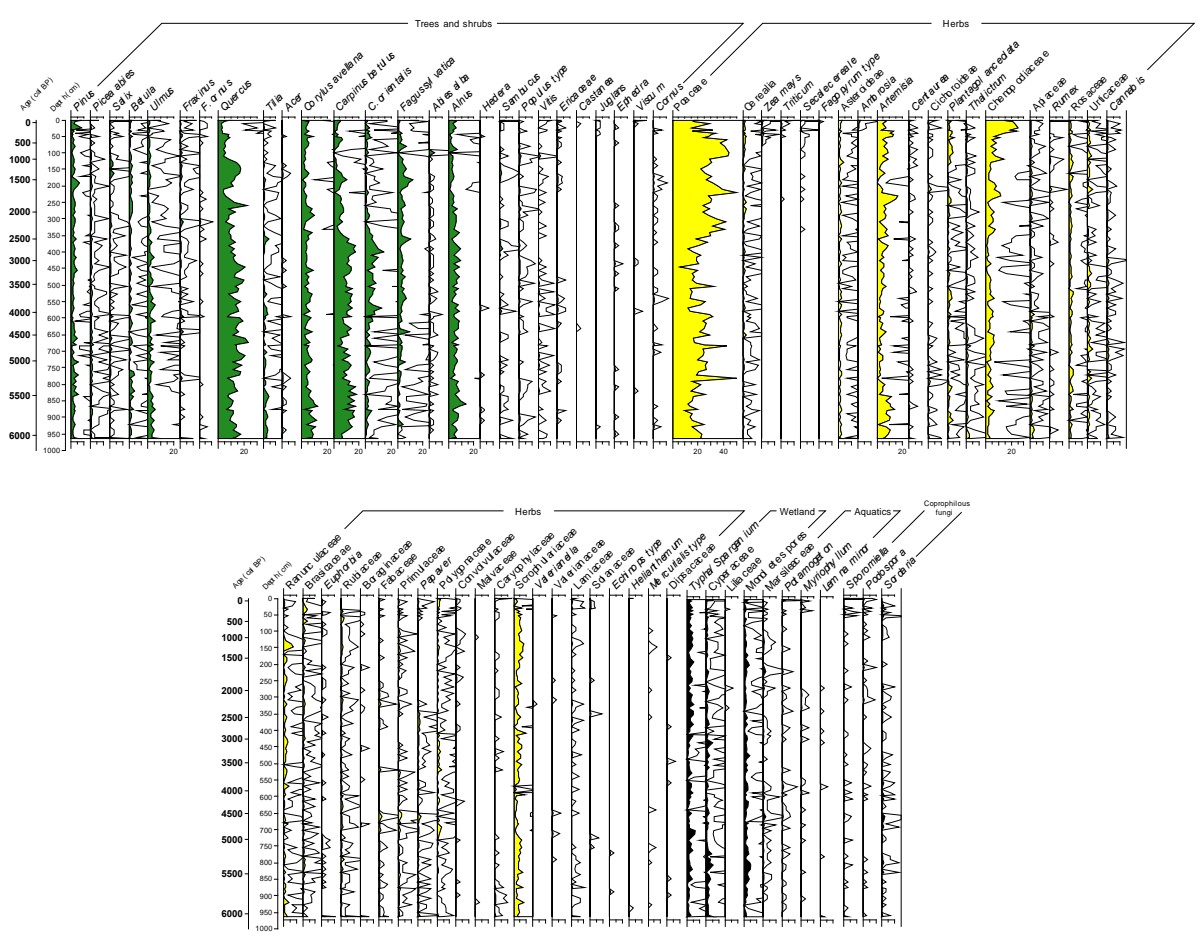

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
