# Peer review of "The transformation of the forest steppe in the lower Danube Plain of south-eastern Europe: 6000 years of vegetation and land use dynamics"

_Biogeosciences, 2020_

## Referee Comment (RC1) · Natalie Schroeter (Referee) · 20 Aug 2020

The manuscript titled "The transformation of the forest steppe in the lower Danube Plain of south-eastern Europe: 6000 years of vegetation and land use dynamic" by Feurdean et al. presents a multi-proxy approach to investigate the history of European forest expansion in the Lower Danube Plain over the last 6000 years. It is a well written and encompasses a detailed comparison to other regions and a comparison to the output of the REVEALS model. Generally, there are no major issues with this manuscript as it is already well organized. I have only minor suggestions and comments which will hopefully help with the data interpretation, particularly of the n-alkane results.

[Figure]

Line 68: It would be interesting to know how far in time human occupation in this area has been detected.

Line 178: "To determine the climate conditions and the predominant vegetation type...". I would suggest to rephrase this sentence and be a little bit more careful with the interpretation value of n-alkanes. n-alkanes are commonly used to distinguish different origins of organic matter sources based on their chain-length and their delta-Deuterium values can help to identify past environmental conditions.

Line 183: N2 subscript

Chapter 3.3 Regime disturbances by fire and herbivores: Since you aim to reconstruct past fires, have you considered analyzing levoglucosan as a proxy for biomass burning?

Line 280: Please explain how shorter chain lengths of n-alkanes correlate to moister climate conditions.

Line 290: Since you discuss climate dynamics in your study region, you should add a short paragraph about the current climate regime/atmospheric systems effecting the studied lake in section 2. Study area.

Line 317: "As climate conditions remained relatively moist during this decline, anthropogenic rather than climatic causes are likely" How did you infer moist conditions?

Line 326: "The proportion of Cerealia is significantly greater in the REVEALS estimate (40%) than in the raw pollen percentages (5%)" Interesting, why is that?

Line 367: "Our quantitative record of vegetation indicates a higher than present tree cover across the landscape of the eastern Lower Danube Plain between 6000 and 4200 cal yr BP with an absolute maxima of 50% (60% raw pollen percentages) between 4200 and 2500 cal yr BP (Fig. 6, see Table 3 for reference and Fig. 1 for locations)." Does the concentration of n-alkanes also increased between 6000 and 4200 cal yr BP?

Line 380: You compare your results to a various other locations. However, did you also consider altitude variations?

Figures

Figure 3: I would suggest to delete the number 850 and possibly 950 from the y axis of the depth [cm] since they are too close to the previous and subsequent depth numbers and thus nearly illegible.

Figure 4: The numbers on the x axis and y axis are a little bit hard to read due to the small font size.

---

## Referee Comment (RC2) · Simon Connor (Referee) · 28 Aug 2020

This manuscript, by Angelica Feurdean and colleagues, is an interesting and professionally executed study of past conditions on the Lower Danube Plain. The authors use a multiproxy approach and quantitative land-cover modelling to address questions about the past extent and dynamics of the forest-steppe ecotone in the Western Black Sea region. The manuscript is very well referenced, contains high quality data and is clearly written. The high temporal resolution of the sampling and the quantitative modelling aspects really make this paper stand out from all others in the region. I agree with the other reviewer (#1) that the manuscript presents no major problems. However, I do have a number of suggestions for further improvement that may improve the manuscript's structure and its interdisciplinary and international appeal.

Specific comments:

1. The Introduction would benefit from some careful restructuring to link the literature review to the research questions more closely. At the moment, the Introduction seems to be presenting many different aims and objectives: "determine lake catchment and in-lake changes", "explore the role of climate, natural and anthropogenic disturbances", test a hypothesis about the naturalness of the landscape, determine whether forests were more moisture-demanding, determine the timing of transformations, determine the ecosystem's sensitivity to climate and anthropogenic impacts, inform decision making about desertification and to test land-cover models. This seems like too many questions for one paper – the authors cannot hope to deliver on all of these with depth and certainty. Indeed, many of these themes are not revisited in the Discussion section. A more focussed introduction would clarify exactly what problems the authors are aiming to (and can) solve. Ideally, the research questions should arise from gaps or uncertainties in the literature.

2. The authors' use of Potential Natural Vegetation (PNV) as a baseline could be more critically assessed. PNVs are problematic since they are static in space and time, while pollen data and REVEALS reconstructions, like the results presented here, show that the past vegetation was spatio-temporally dynamic. There is an excellent paper exploring the mismatches between PNV and REVEALS in Czechia (Abraham et al. 2016, Preslia 88: 409-434) and I encourage the authors to consult it and other papers on the topic (e.g. Chiarucci et al. 2010, J. Veg. Sci. 21: 1172-1178; Rull 2015, J. Veg. Sci. 26: 603-607). My feeling is that the manuscript would be stronger if the authors reduced their reliance on the PNV map and instead used the REVEALS reconstruction as a test for PNV accuracy. This is important because PNV is commonly used as a baseline for restoration and it could be influencing current conservation efforts. The paper shows that there are several possibilities for the vegetation of the site – various

types of forest-steppes, steppes and agro-pastoral landscapes, which are dynamic in space and time. This would make for much richer and more interesting conclusions.

3. Given the carbonate-rich geology and the fact that mostly shells were dated, it would be useful to briefly discuss potential reservoir effects. Reservoir effects will not change the modelling results, but may introduce some uncertainty about the timing of the major transitions identified in the land-cover reconstructions and the interpretations of vegetation change as being forced by climate.

4. A surprising omission in the manuscript is the aquatic pollen taxa. These taxa might help better interpret the n-alkane results, providing additional proxy for lake hydrological conditions. The authors' interpretations of anoxia and lake level change may find stronger support with the addition of aquatic pollen. The extent to which aquatic vegetation influenced the n-alkanes signal could also be explored.

5. The REVEALS modelling in this paper is very comprehensive. One small doubt concerns Carpinus orientalis. In my experience, this 'tree' is very often a shrub in forest-steppe landscapes. Is it correct to call it a 'forest' taxon in this region? Is it possible that the increase in C. orientalis around 4200 cal. yr BP represents a scrub expansion or the abandonment of coppicing? These changes may even relate to shifting land-use at the Neolithic–Bronze Age transition, a topic that would benefit from further exploration in the Discussion.

6. Overall, I found this a very interesting manuscript and a novel contribution to the literature on the antiquity and dynamics of temperate grasslands.

Technical comments by line number:

31 – place a comma after "taxa" 32 – "Maximum tree cover... between 4200 and 2500" here, but "greatest tree cover... between 6000 and 2500" in line 30... this seems like a mistake since the ranges overlap 35 – "mid-Holocene forest maximum" seems to refer to conditions from 4200-2500 cal. yr BP, which is usually termed late Holocene.

See also comments about the use of "forest", line 305. 38 – "highlighting recurring anthropogenic pressure" – is this indeed cyclical, or due to more-or-less continuous anthropogenic pressure? 39 – "was in between that in" – consider "falls in between that of" 42 – "reflects" – change to "reflect" to agree with woodlands 44 – the comment about pollen preservation seems to indicate that this pollen record may be adversely affected by taphonomy… is that the case? Perhaps taphonomy could be addressed elsewhere in the paper? 47 – delete "The" at start of sentence 49-51 – this sentence sounds like it concerns the present-day environment (i.e. ecological studies), but all the references are palaeo studies. More precise wording would avoid misleading readers, or modern ecological studies could be incorporated to increase the interdisciplinarity of the manuscript 54 – the definition of "lowlands" in this paper is not very clear, since it seems to imply that lowlands are "drier" than "mesic areas of Europe". A more precise term like "steppic grasslands" or even a map showing the current extent (not potential extent) of the ecosystem might help, since readers could easily confuse the steppic lowlands they are referring to with the lowlands of the Netherlands, for example, which are certainly quite different. 65 – the idea of grasslands being richer than rainforests seemed surprising, but the original reference states "at small spatial scales vascular plant diversity of certain European grasslands even exceeds tropical rainforests". It would be good to specify what scale is meant here, since 'alpha diversity' is a vague term encompassing a large range of spatial scales (Whittaker et al. 2001, J. Biogeog. 28: 453-470). 67 – this sentence refers to temperate grasslands (it may be different for tropical grasslands) 71 – unclear what you mean by "cultural rise" – what is it and what is the evidence for it? 87 – remove comma after "although". I found this sentence difficult to understand. Are you saying that recent land-cover models tend to support the pollen-based reconstructions, whereas the old ones didn't? 92 – this sentence is very important in the paper, but is very complex and methodological. I feel as though it would be more logical to have it appear after the aim and research questions, rather than before. 96 – these are very good research questions and would benefit from stronger links to the preceding introductory material. Why is it important to know what

is "natural" here? Is that a current concern for biodiversity conservation in the region? Why would we expect moisture-demanding taxa in past forests? Has this been hypothesised on some basis? There seems to be a word missing in "When did this area undergo the most marked land cover"... land-cover change, perhaps? 103 – what is the hypothesis mentioned here? How can it be assessed using pollen data which are poorly taxonomically resolved for many grassland taxa? 105 – please explain these desertification phenomena earlier in the introduction to provide context 114 – what is the source of the "excessively temperate continental" category? Could you give a Köppen classification as well? 118 – comma after "eroded" 120 – it's unclear to me why the potential natural vegetation is being presented here, given that the paper aims to reconstruct the past vegetation and land cover. I feel that PNV should be a model to be tested, rather than stated as a fact, as it is here. 126 – the pollen source area is stated as being 20 km. How was this determined? 162 – what is the minimum pollen sum? 177 – it would be good to justify the statement that n-alkanes reflect climate conditions here – what is the evidence? 199 – this statement about trees may need revision to specify deciduous trees or angiosperms, since n-alkanes appear to be less successful in detecting coniferous trees (Diefendorf et al. 2015. Geochim. Cosmochim. Acta 170: 145-156) 220-225 – please provide a reference justifying Zr and Fe:Mn as proxies for erosion and anoxia 273 – the subtitle here is confusing and unclear. Are you trying to say it's about both periods or the transition from one to the other? Perhaps "Transition from forest steppe (6000-4200 cal. BP) to maximum tree cover (4200-2500 cal. BP)" would be clearer? Also, can we be certain C. orientalis was a 'tree'? 276 – change "open woodland forest" to "open woodland" 279 – change "shrub" to "shrubs" (or make grasses and forbs singular) 280 – change "Coeval to" to "Coeval with" 282 – is it possible to disentangle the climatic and vegetation signals in the n-alkanes, or is it not possible to say whether they were caused by one or the other? Perhaps the pollen record suggests that vegetation thickening was the main cause of the n-alkanes variations? Vegetation thickening might be linked to declining fire activity, changes in agricultural practices (cropping, grazing) and/or climatic drivers. 286 – how are aquatic plants reflected in the n-alkane record? The absence of aquatic/wetland taxa (e.g. Cyperaceae) in the pollen record (Appendix A1) is striking – were aquatics not counted? 288 – is it possible that there was a higher lake level when this lake is connected hydrologically to the Danube? This would imply the Danube also having a higher level, wouldn't it? Could the potential anoxia be linked to lake trophic status or to aquatic vegetation? 305 – the term "forest cover" is a little bit misleading as the authors' reconstructions do not show the existence of a forest, but of a forest-steppe. There are several instances through the text where this confusion could arise, including line 34 of the Abstract. In the context of forest-steppe, I suggest "tree cover" rather than "forest cover" and perhaps "steppe expansion" rather than "forest loss". 311 – change "impact of" to "impact on" 325 – remove the "." after "Plantago" 329 – the mention of uncertainties is welcome; however, the authors could help the reader to better understand the implications of the uncertainties... do the interpretations change if we consider these uncertainties, or are the uncertainties minor? 353 – replace "modelled-based" with "model-based" 392 – unclear sentence 398 – change "extend" to "extent" 399 – this is an interesting idea about the climatic suitability of the region for forest, though it is maintained relatively open through disturbances like cropping and grazing – a 'plagioclimax'. Perhaps this idea could be expanded a bit more, given its relevance to landscape restoration and conservation, as well to understanding human interactions with the biota of the region 403 – delete "the" before "SE Europe" 411 – comma after "millennia" 412 – the idea that deforestation contributed to aridification would benefit from some discussion and justification in the Discussion section, not just in the Conclusion. 416 – there is a slight inconsistency in the argument here about n-alkanes. The authors say that they track the vegetation changes in the pollen, so that makes them a reliable indicator of past vegetation change, but earlier in the paper the n-alkanes are interpreted as a climate proxy. Is it that n-alkanes are a proxy for climate directly, or are they, like pollen, a proxy for vegetation, which can be influenced by climate? Readers like myself will be grateful for the additional explanation! 418 – please help the reader to understand how this single record might be able to be used to test land-cover models – this would add greatly

to the application and relevance of the study 419 – what is meant by "an earlier impact than in the reconstructions is also true"? This issue is not really discussed in the paper and seems not to really relate to the material presented. Perhaps after the research questions are refined, the conclusion could be restructured slightly to address those? 425 – replace "design" with "designed" 431 – replace "grating" with "granting" (also line 433) Fig. 5 – please add some indication of the pollen zones from Fig. 4 Table 2 – check spelling of Plantago lanceolata Appendix A1 – please add wetland/aquatic taxa to the pollen diagrams

---

## Author Comment (AC1) · 18 Sep 2020

18.09.2020, Frankfurt am Main, feurdean@em.uni-frankfurt.de

The transformation of the forest steppe in the lower Danube Plain of south-eastern Europe: 6000 years of vegetation and land use dynamic" by Angelica Feurdean et al. Simon Connor (Referee) simon.connor@anu.edu.au

This manuscript, by Angelica Feurdean and colleagues, is an interesting and professionally executed study of past conditions on the Lower Danube Plain. The authors

use a multiproxy approach and quantitative land-cover modelling to address questions about the past extent and dynamics of the forest-steppe ecotone in the Western Black Sea region. The manuscript is very well referenced, contains high quality data and is clearly written. The high temporal resolution of the sampling and the quantitative modelling aspects really make this paper stand out from all others in the region. I agree with the other reviewer (#1) that the manuscript presents no major problems. However, I do have a number of suggestions for further improvement that may improve the manuscript's structure and its interdisciplinary and international appeal.

R: Many thanks for your positive appreciation of our work.

Specific comments: 1.The Introduction would benefit from some careful restructuring to link the literature review to the research questions more closely. At the moment, the Introduction seems to be presenting many different aims and objectives: "determine lake catchment and in-lake changes", "explore the role of climate, natural and anthropogenic disturbances", test a hypothesis about the naturalness of the landscape, determine whether forests were more moisture-demanding, determine the timing of transformations, determine the ecosystem's sensitivity to climate and anthropogenic impacts, inform decision making about desertification and to test land-cover models. This seems like too many questions for one paper – the authors cannot hope to deliver on all of these with depth and certainty. Indeed, many of these themes are not revisited in the Discussion section. A more focussed introduction would clarify exactly what problems the authors are aiming to (and can) solve. Ideally, the research questions should arise from gaps or uncertainties in the literature.

R: We largely agree with this point. As this study is the first mid to late Holocene multi-proxy palaeoenvironmental and palaeovegetation reconstruction for this region, we highlighted all the main questions we felt we could answer with the current datasets. However, our aims are targeted at understanding the past vegetation composition and dynamics and their drivers (climate, humans and fire). In the revised version of the paper, we will streamline the aims to ensure a better fit with the themes addressed in

the Discussion.

2.The authors' use of Potential Natural Vegetation (PNV) as a baseline could be more critically assessed. PNVs are problematic since they are static in space and time, while pollen data and REVEALS reconstructions, like the results presented here, show that the past vegetation was spatio-temporally dynamic. There is an excellent paper exploring the mismatches between PNV and REVEALS in Czechia (Abraham et al. 2016, Preslia 88: 409-434) and I encourage the authors to consult it and other papers on the topic (e.g. Chiarucci et al. 2010, J. Veg. Sci. 21: 1172-1178; Rull 2015, J. Veg. Sci. 26: 603-607). My feeling is that the manuscript would be stronger if the authors reduced their reliance on the PNV map and instead used the REVEALS reconstruction as a test for PNV accuracy. This is important because PNV is commonly used as a baseline for restoration and it could be influencing current conservation efforts. The paper shows that there are several possibilities for the vegetation of the site – various types of forest-steppes, steppes and agro-pastoral landscapes, which are dynamic in space and time. This would make for much richer and more interesting conclusions.

R: This is a pertinent point and one that can be addressed with pollen records. Our study does not only rely on the accuracy of PNV map to determine the natural vegetation in the region but uses the pollen-based vegetation reconstruction to determine how much the current vegetation diverges from what should be there under the same climatic conditions. As this point appears to not have come across clearly in the submitted manuscript, we will work to incorporate the reviewer's suggestions and additionally highlight the mismatch between the vegetation composition from PNV map and the pollen-based vegetation reconstruction. We also thank the reviewer for the useful references he has provided.

3.Given the carbonate-rich geology and the fact that mostly shells were dated, it would be useful to briefly discuss potential reservoir effects. Reservoir effects will not change the modelling results, but may introduce some uncertainty about the timing of the major transitions identified in the land-cover reconstructions and the interpretations of vegetation change as being forced by climate.

R: We have determined that the hard water effect is about 1000 years and discuss the timing of major environmental changes assuming ages 1000 years younger than would have been calculated without taking into account this dating limitation (see S2). In the revised discussion we will also refer to the timing of major vegetation transition according to the uncorrected age depth.

4. A surprising omission in the manuscript is the aquatic pollen taxa. These taxa might help better interpret the n-alkane results, providing additional proxy for lake hydrological conditions. The authors' interpretations of anoxia and lake level change may find stronger support with the addition of aquatic pollen. The extent to which aquatic vegetation influenced the n-alkanes signal could also be explored.

R: Thank you for pointing this out. The pollen percentages of aquatic taxa and Cyperaceae do not show a clear pattern in their composition and abundance over time to provide any additional information on changes in hydrological conditions. However, in the revised paper we will present these taxa and attempt to link their dynamics to that of n-alkane variations.

5. The REVEALS modelling in this paper is very comprehensive. One small doubt concerns Carpinus orientalis. In my experience, this 'tree' is very often a shrub in forest-steppe landscapes. Is it correct to call it a 'forest' taxon in this region? Is it possible that the increase in C. orientalis around 4200 cal. yr BP represents a scrub expansion or the abandonment of coppicing? These changes may even relate to shifting land-use at the Neolithic–Bronze Age transition, a topic that would benefit from further exploration in the Discussion.

R: We have referred to Carpinus orientalis as a forest taxon based on its percentages and consistent presence in the uppermost canopy levels in our vegetation surveys of/within forest stands in the area. Upon a more thorough and linked investigation of our archaeological and palynological records, we will integrate our findings regarding

the dynamics of C. orientalis in the revised version of the manuscript.

In addition to the general comments, we found most of specific comments very useful and they will help us to improve the quality of the paper. We will therefore make every effort to incorporate the majority of them into the revised version of the paper.

Kind regards, Angelica on behalf of the co-authors

---

## Author Comment (AC2) · 12 Oct 2020

Natalie Schroeter (Referee) nschroet@bgc-jena.mpg.de The manuscript titled "The transformation of the forest steppe in the lower Danube Plain of south-eastern Europe: 6000 years of vegetation and land use dynamic" by Feurdean et al. presents a multi-proxy approach to investigate the history of European forest expansion in the Lower Danube Plain over the last 6000 years. It is a well written and encompasses a detailed comparison to

other regions and a comparison to the output of the REVEALS model. Generally, there are no major issues with this manuscript as it is already well organized. I have only minor suggestions and comments which will hopefully help with the data interpretation, particularly of the n-alkane results.

R: Thank you for the positive appreciation of our work.

Line 68: It would be interesting to know how far in time human occupation in this area has been detected.

R: We have revised this paragraph to fit to the story of Potential Natural Vegetation as requested by Rev 2- (see Rev 2 point 2 and lines 92, 96). It now reads "Since the Lower Danube Plain represents one of the oldest areas of continuous human occupation from the Neolithic onwards i.e., 8000 cal yr BP ( http://ran.cimec.ro; Balasescu and Radu, 2004; Weininger et al., 2009; Wunderlich et al., 2012; Nowacki et al., 2019) and one of the most important agricultural areas in Europe (European Environmental Agency, 2016), the current vegetation is likely very different from its natural state, though exactly how different is not known".

Line 178: "To determine the climate conditions and the predominant vegetation type. . .". I would suggest to rephrase this sentence and be a little bit more careful with the interpretation value of n-alkanes. n-alkanes are commonly used to distinguish different origins of organic matter sources based on their chain-length and their delta Deuterium values can help to identify past environmental conditions.

R: We have rephrased this sentence to more accurately show that the chain-length of n-alkanes are commonly used to distinguish among sources of organic matter. The relation between the n-alkanes chain length and the type of organic material and plant source is also detailed in 3.2.2. Additionally, to accommodate Rev 2 suggestion (l. 286) on the occurrence of n-alkanes of aquatic origin, we have added Paq =aquatic index, which quantifies the abundance of submerged vascular macrophyte. This subsection now reads :"3.2.2 Leaf wax n-alkanes based vegetation reconstruction. To determine

the source of organic matter and the predominant vegetation type (Eglinton and Calvin, 1967; Ficken et al., 2000; Diefendorf et al. 2015), we measured the concentration of higher-plant derived n-alkane homologues of 60 sediment samples selected along the composite core. n-Alkanes are an integral part of higher-plant leaf epicuticular waxes, highly resistant to degradation and among the most stable lipid components of the protective waxes coating terrestrial plant leaves (Eglinton and Eglinton, 2008; Sachse et al., 2012). They are commonly used to distinguish sources of organic matter based on their chain-length (….). In this study, we calculated the ratio of straight-chain n-alkanes of different chain (homologues) lengths as these have been previously used as proxies for the relative contribution of various types of plants in lacustrine sediments (e.g. Ficken et al., 2000; Zhou et al., 2005). Average chain length (ACL) is an indicator of the relative abundance of short (C16-C20) vs long chain n-alkanes and may be linked to the predominance of higher taxonomic plants over lower taxonomic plants including algae (Ficken et al., 2000; Eglinton and Calvin, 1967). Within the long chain n-alkanes, the abundance of n-alkanes with n-C31 and n-C33 may be indicative of grass predominance, whereas n-C27 and n-C29 may indicate a predominantly tree covered landscape (Aichner et al., 2010; Meyers, 2003). The aquatic index (Paq) quantifies the abundance of submerged and floating vascular macrophytes, which are characterised by medium chain length n-alkanes (n-C23 and n-C25), relative to emergent aquatic and terrestrial plant types that are characterised by long chain n-alkanes (Ficken et al., 2000). It should be noted that n-alkanes are less successful in detecting coniferous than angiosperms (Diefendorf et al. 2015) and that some overlap within the mid chain length alkanes n-C23 and n-C25 is possible (Aichner et al., 2010; Meyers, 2003)".

Line 183: N2 subscript Chapter 3.3 Regime disturbances by fire and herbivores: Since you aim to reconstruct past fires, have you considered analyzing levoglucosan as a proxy for biomass burning?

R: We could not reliably detect levoglucosan in any of the trial samples. Levoglucosan peaks detected in a few samples were very small relative to the high background,

meaning a high degree of uncertainty in the analysis result.

Line 280: Please explain how shorter chain lengths of n-alkanes correlate to moister climate conditions.

R: To avoid confusion (see also comment l.282 of Rev 2), we have removed the interpretation of the shorter chain length as a direct proxy for moist conditions and retained only that the (C27+C29)/(C33/C31) ratio primary reflect changes in major vegetation type. However, indirectly, increases in the (C27+C29)/(C33/C31) ratio typical for the dominance of trees may indicate wetter conditions, as trees required higher amount of moisture than the grasslands, especially steppic one. This sentence now reads: "Coeval with the maximum extent in tree cover, the n-alkanes were dominated by the shorter chain lengths (ALC) and a higher n(C27+C29)/(C33/C31) ratio, indicative of increased contribution of tree-derived n-alkanes (Meyers, 2003; Aichner et al., 2010). However, the concentration of n-alkane varied with that of detrital elements (high between 6000 and 4200 cal yr BP; low between 4200-25000 cal yr BP; Fig.3) suggesting their enhanced deliver into the lake by runoff of floods".

2. Study area. 290: Since you discuss climate dynamics in your study region, you should add a short paragraph about the current climate regime/atmospheric systems effecting the studied lake in section

R: This and the suggestions of Rev2_114 we have revised the information about the climate type in this region. This reads: "The climate in the study region is wet-warm temperate continental (Koppen-Geigger class Dfa), also termed excessive with the prevalence of harsh winters and hot summers (Posea et al., 2005), due to the influence of air masses from continental Asia. The mean annual temperature is of ca. 11°C, mean January temperature of ca. -1 °C and a mean summer temperature 25°C. Annual precipitation is about 400 mm (Adamclisi Meteo station)."

Line 317: "As climate conditions remained relatively moist during this decline, anthropogenic rather than climatic causes are likely" How did you infer moist conditions?

R: We have removed the climatic hypothesis from this place, and only discussed pollen-based evidence of anthropogenic land cover changes. In the second part of this sub-chapter (5.2), we discuss local climate conditions based on P aq, and geochemistry as well as published regional climatic records and conclude that climate conditions remained relatively moist during this decline in tree cover, therefore anthropogenic impact was likely the main cause of tree cover loss.

Line 326: "The proportion of Cerealia is significantly greater in the REVEALS estimate (40%) than in the raw pollen percentages (5%)" Interesting, why is that?

R: To accommodate this and at the suggestion of Rev2 (line 329) we have introduced a paragraph discussing the implication of uncertainties in the REVEALS model on the pollen-based vegetation reconstruction. This reads: "Our Cerealia pollen includes Triticum, Zea, Hordeum and Secale cereale, for which we have used productivity estimates obtained from calibration of local surface pollen samples with vegetation inventory (Grindean et al., 2019). These productivity estimates are considerably lower than the average for Europe (0.22 vs. 1.85; Mazier et al., 2012), which is the main cause for the high proportion of Cerealia cover reconstructed by REVEALS and for the difference in outcome from this study and elsewhere in Europe. The crop species included in Cerealia also varied among regions and with time, which may also introduce differences when applying the PPEs for landscape reconstruction back in time. Lastly, the occurrence of wild grass species with pollen that fall in the Cerealia pollen type (all Poaceae grains larger than 40 $\mu$m), may have led to an overestimation of proportion of Cerealia at certain times in the past".

Line 367: "Our quantitative record of vegetation indicates a higher than present tree cover across the landscape of the eastern Lower Danube Plain between 6000 and 4200 cal yr BP with an absolute maxima of 50% (60% raw pollen percentages) between 4200 and 2500 cal yr BP (Fig. 6, see Table 3 for reference and Fig. 1 for locations)." Does the concentration of n-alkanes also increased between 6000 and 4200 cal yr BP?
R: We have extended the discussion on the n-alkane concentration See reply l. 280.

You compare your results to a various other locations. However, did you also consider altitude variations?

R: All sites are located at low elevation (less than 300 m asl). To illustrate this, in Table 3, we have added elevation of each site.

Figures Figure 3: I would suggest to delete the number 850 and possibly 950 from the y axis of the depth [cm] since they are too close to the previous and subsequent depth numbers and thus nearly illegible.

R: For a better readability, we amended depths on this figure.

Figure 4: The numbers on the x axis and y axis are a little bit hard to read due to the small font size.

R: For a better readability, we enlarged the numbers on this figure.

---

## Author Comment (AC3) · 12 Oct 2020

BGD Interactive comment Biogeosciences Discuss., https://doi.org/10.5194/bg-2020-239-RC2, 2020[©] Author(s) 2020. This work is distributed under the Creative Commons Attribution 4.0 License. Interactive comment on "The transformation of the forest steppe in the lower Danube Plain of south-eastern Europe: 6000 years of vegetation and land use dynamic" by Angelica Feurdean et al. Simon Connor (Referee) simon.connor@anu.edu.au

This manuscript, by Angelica Feurdean and colleagues, is an interesting and professionally executed study of past conditions on the Lower Danube Plain. The authors use a multiproxy approach and quantitative land-cover modelling to address questions about the past extent and dynamics of the forest-steppe ecotone in the Western Black Sea region. The manuscript is very well referenced, contains high quality data and is clearly written. The high temporal resolution of the sampling and the quantitative modelling aspects really make this paper stand out from all others in the region. I agree with the other reviewer (#1) that the manuscript presents no major problems. However, I do have a number of suggestions for further improvement that may improve the manuscript's structure and its interdisciplinary and international appeal.

R: Many thanks for your positive appreciation of our work. Please note that this is our extensive response to the review compared to our response dated 18.09.2020.

Specific comments: 1.The Introduction would benefit from some careful restructuring to link the literature review to the research questions more closely. At the moment, the Introduction seems to be presenting many different aims and objectives: "determine lake catchment and in-lake changes", "explore the role of climate, natural and anthropogenic disturbances", test a hypothesis about the naturalness of the landscape, determine whether forests were more moisture-demanding, determine the timing of transformations, determine the ecosystem's sensitivity to climate and anthropogenic impacts, inform decision making about desertification and to test land-cover models. This seems like too many questions for one paper – the authors cannot hope to deliver on all of these with depth and certainty. Indeed, many of these themes are not revisited in the Discussion section. A more focussed introduction would clarify exactly what problems the authors are aiming to (and can) solve. Ideally, the research questions should arise from gaps or uncertainties in the literature.

R: As this study is the first mid to late Holocene multi-proxy palaeoenvironmental and palaeovegetation reconstruction for this region, we highlighted all the many questions we felt we could answer with the current datasets. In the revised version of the paper, we streamlined the aims to ensure a better fit with the themes addressed in the Discussion, i.e., understanding the past vegetation composition and dynamics and their

drivers (climate, humans and fire). The aims now read: "Here, we explore the long-term vegetation dynamics of the Lower Danube Plain landscapes and the competing driving forces (climate, fire and anthropogenic impact). More specifically, we address the following research questions: i) Is forest steppe the natural vegetation type of the Lower Danube Plain under climatic conditions similar to those in the present? ii) Has the tree cover been more extensive or dominated by other tree taxa in the past? iii) When did this area undergo the most marked land cover and land use changes and was this transformation continuous in time? This study is built on a pollen-based quantitative vegetation reconstruction (REVEALS model), along with records of long-chain higher plant-wax n-alkanes, charcoal, coprophilous fungi and geochemistry from a sedimentary sequence from Lake Oltina, south-eastern Romania. This is the first pollen based quantitative land cover estimate in the forest steppe of south eastern Europe and allows the hypothesis of the naturalness of forest steppe ecosystems in this region as well as its sensitivity to climate and anthropogenic impact to be critically tested". 100.

2.The authors' use of Potential Natural Vegetation (PNV) as a baseline could be more critically assessed. PNVs are problematic since they are static in space and time, while pollen data and REVEALS reconstructions, like the results presented here, show that the past vegetation was spatio-temporally dynamic. There is an excellent paper exploring the mismatches between PNV and REVEALS in Czechia (Abraham et al. 2016, Preslia 88: 409-434) and I encourage the authors to consult it and other papers on the topic (e.g. Chiarucci et al. 2010, J. Veg. Sci. 21: 1172-1178; Rull 2015, J. Veg. Sci. 26: 603-607). My feeling is that the manuscript would be stronger if the authors reduced their reliance on the PNV map and instead used the REVEALS reconstruction as a test for PNV accuracy. This is important because PNV is commonly used as a baseline for restoration and it could be influencing current conservation efforts. The paper shows that there are several possibilities for the vegetation of the site – various types of forest-steppes, steppes and agro-pastoral landscapes, which are dynamic in space and time. This would make for much richer and more interesting conclusions.
R: This is a pertinent point. As this point appeared to not have come across clearly, in the revised version, we have incorporated the reviewer's suggestions and more specifically highlight the issues of using PNV as a baseline for vegetation composition. This now reads: According to Bohn et al. (2003), the potential natural vegetation (PNV) in the study area, the easternmost part of the Lower Danube Plain, also known as the Southern Dobrogea Plateau, is forest steppe, i.e., woodland patches within a matrix of graminoid and forb dominated communities. It borders steppe grasslands i.e., treeless vegetation dominated by graminoids and forbs to the east and thermophilous mixed deciduous broadleaf forests to the west. The forest steppe and steppe vegetation extend over 6000 km along an east-west gradient across Eurasia (Fig. 1) under climate conditions delimited by a ca. 2 month long late-summer drought for the forest steppe zone and 4–6 months for the steppe (Walter, 1974). However, according to the management plan for the region (Planul de Management, 2016), there are currently few patches of natural steppe vegetation preserved in the Southern Dobrogea Plateau as most have become ruderal steppe. Since the Lower Danube Plain represents one of the oldest areas of continuous human occupation from Neolithic onwards i.e., 8000 cal yr BP (http://ran.cimec.ro; BălăÈŹescu and Radu, 2004; Weininger et al., 2009; Wunderlich et al., 2012; Nowacki et al., 2019) and one of the most important agricultural areas in Europe (European Environmental Agency, 2016), the current vegetation is likely very different from its natural state, though exactly how different is not known. The use of PNV as a baseline for natural vegetation has been found to be problematic as it considers only climax vegetation i.e., the final stage of an ecological succession, and assumes vegetation to remain static in space and time (Chiarruci, et al., 2010; Jackson et al., 2013; Abraham et al., 2014; Rull 2015). PNV also fails to consider the vital role of natural disturbances such as fire and herbivores and their impacts on vegetation succession (Chiarruci, et al., 2010; Feurdean et al., 2018), leading to unrealistic vegetation reconstruction even in the absence of disturbance by humans (Jackson et al., 2013). These are the reasons why most areas that are currently covered by grasslands or open woodlands in Central Eastern Europe are given as naturally dominated by deciduous broadleaf forest or mixed coniferous and broadleaf forest in PNV (Feurdean et al., 2018) and the representation of pioneer trees in PNV is much lower compared to the pollen-based Holocene vegetation estimates (Abraham et al., 2014). Inaccurate identification of natural vegetation types can also lead to inappropriate decision in terms of conservation practices and policies. For example, the worldwide reforestation used PNV to identify opportunities for landscape restoration to mitigate climate change in areas where the climate can sustain forest (http://www.wri.org/applications/maps/flr-atlas). This approach is threatening grassland ecosystems as such policies are based on the false assumption that most grasslands are man-made and ignore the very high richness of grasslands at smaller spatial scale (Whittaker et al. 2001), as well as their unique cultural significance (Dengler et al., 2014)." 55-80.

3.Given the carbonate-rich geology and the fact that mostly shells were dated, it would be useful to briefly discuss potential reservoir effects. Reservoir effects will not change the modelling results, but may introduce some uncertainty about the timing of the major transitions identified in the land-cover reconstructions and the interpretations of vegetation change as being forced by climate

R: We have determined that the hard water effect is about 1000 years and discuss the timing of major environmental changes assuming ages 1000 years younger than would have been calculated without taking into account this dating limitation (see S2). In the revised discussion we also discusse to the timing of major deforestation and the link with archaeological evidences and simulated vegetation suing both original and adjusted age.

4. A surprising omission in the manuscript is the aquatic pollen taxa. These taxa might help better interpret the n-alkane results, providing additional proxy for lake hydrological conditions. The authors' interpretations of anoxia and lake level change may find stronger support with the addition of aquatic pollen. The extent to which aquatic vegetation influenced the n-alkanes signal could also be explored.

R: We have included aquatic and wetland taxa into the pollen diagram (Appendix A1) and discuss potential connection with hydrological conditions in the revised version of the paper.

5. The REVEALS modelling in this paper is very comprehensive. One small doubt concerns Carpinus orientalis. In my experience, this 'tree' is very often a shrub in forest-steppe landscapes. Is it correct to call it a 'forest' taxon in this region? Is it possible that the increase in C. orientalis around 4200 cal. yr BP represents a scrub expansion or the abandonment of coppicing? These changes may even relate to shift-ing land-use at the Neolithic–Bronze Age transition, a topic that would benefit from further exploration in the Discussion.

R: According to the European Atlas of Forest Tree Species (2016), Carpinus orientalis is described as a small tree or shrub, and as a tree species native to southeast Europe, the Pontic region and western Asia. In Bulgaria, Dodev et al. (2020) Austrian Journal of Forests Science, 137, 23-42, 2020, defined oriental hornbeam forests as forest stands with 50 % or more of C. orientalis in the tree composition. Therefore, we have referred to C. orientalis as a forest taxon based on its percentages and consistent presence in the uppermost canopy levels in our vegetation surveys of/within forest stands in the area. The increase of Carpinus orientalis around 4200 cal BP at the expense of Quercus is concurrent with the expansion of Fagus sylvatica and C. betulus and a low abundance of other local shrub taxa (Appendix A1), thus suggesting its expansion as a tree not scrub expansion. Archaeological findings from the area are very scarce for the first half (4200-3400 cal yr BP) of the maximum tree cover, corresponding to the early and mid-Bronze Age ((http://ran.cimec.ro;). The decrease of Quercus could indicate deforestation, although the low abundance of anthropogenic pollen indicators (Cerealia) and coprophilous fungi suggest logging was at a small scale, probably for individual households.

Technical comments by line number:

– place a comma after "taxa" 32 – "Maximum tree cover. . . between 4200 and 2500" here, but "greatest tree cover. . . between 6000 and 2500" in line 30. . . this seems like a mistake since the ranges overlap

R: We have slightly altered this sentence to show that there was a higher tree cover 6000-2500 cal yr BP, with absolute maxima between 4200-2500 cal yr BP. "We found a greater tree cover, composed of xerothermic (Carpinus orientalis and Quercus) and temperate (Carpinus betulus, Tilia, Ulmus and Fraxinus) tree taxa, between 6000 and 2500 cal yr BP. Maximum tree cover (~50%) occurred between 4200 and 2500 cal yr BP at a time of wetter climatic conditions.

– "mid-Holocene forest maximum" seems to refer to conditions from 4200-2500 cal. yr BP, which is usually termed late Holocene. Line 34-See also comments about the use of "forest", line 305. In the context of forest-steppe, I suggest "tree cover" rather than "forest cover" and perhaps "steppe expansion" rather than "forest loss".

R: We have corrected mid with late Holocene. Additionally, we have changed forest cover with tree cover through the manuscript and tree loss and steppe expansion in the Abstract. This sentence reads "Tree loss // therefore steppe expansion, was under way by 2500 yr BP (Iron Age) with REVEALS estimates indicating a fall to ~20% tree cover from the late Holocene forest maximum linked to".

– "highlighting recurring anthropogenic pressure" – is this indeed cyclical, or due to more-or-less continuous anthropogenic pressure?

R. Changed. A sparse tree cover, with only weak signs of forest recovery, then became a permanent characteristic of the Lower Danube Plain, highlighting more or less continuous anthropogenic pressure.

– "was in between that in" – consider "falls in between that of" R: Changed to: "The timing of anthropogenic ecosystem transformation here (2500 cal yr BP) falls in between that in central eastern (between 3700 and 3000 cal yr BP) and eastern (after cal yr BP) Europe".

– "reflects" – change to "reflect" to agree with woodlands

R: Changed to reflect.

– the comment about pollen preservation seems to indicate that this pollen record may be adversely affected by taphonomy. . . is that the case? Perhaps taphonomy could be addressed elsewhere in the paper?

R: We did not run into the problem of pollen preservation at Lake Oltina. We remove this sentence from the abstract and preserve it at the conclusion to highlight that n-alkane could accurately track changes in the main vegetation types i.e., tree-grass cover in this region.

– delete "The" at start of sentence 49-51 – this sentence sounds like it concerns the present-day environment (i.e. ecological studies), but all the references are palaeo studies. More precise wording would avoid misleading readers, or modern ecological studies could be incorporated to increase the interdisciplinarity of the manuscript 54 – the definition of "lowlands" in this paper is not very clear, since it seems to imply that lowlands are "drier" than "mesic areas of Europe". A more precise term like "steppic grasslands" or even a map showing the current extent (not potential extent) of the ecosystem might help, since readers could easily confuse the steppic lowlands they are referring to with the lowlands of the Netherlands, for example, which are certainly quite different.

R: We have split this sentence in two; the first refers to contemporary studies the second to past trends. It reads: "Lowland ecosystems (mesic and steppic grasslands, woodlands, etc.) provide an array of provisioning (e.g. crops, grazed areas, wood) and regulating services (e.g. soil protection; European Environmental Agency, 2016). However, in comparison to the mountainous areas of central eastern Europe, lowland ecosystems, especially steppic grasslands have been more strongly impacted by human activities (Magyari et al., 2010. . .)".

– the idea of grasslands being richer than rainforests seemed surprising, but the original reference states "at small spatial scales vascular plant diversity of certain European grasslands even exceeds tropical rainforests". It would be good to specify what scale is meant here, since 'alpha diversity' is a vague term encompassing a large range of spatial scales (Whittaker et al. 2001, J. Biogeog. 28: 453-470). 67 – this sentence refers to temperate grasslands (it may be different for tropical grasslands)

R: Added. It reads:" This approach is threatening grassland ecosystems as such policies are based on the false assumption that most grasslands are man-made and ignore the very high richness of grasslands at smaller spatial scale (Whittaker et al. 2001), as well as their unique cultural significance (Dengler et al., 2014).

– unclear what you mean by "cultural rise" – what is it and what is the evidence for it?

R: We have revised this paragraph. It reads: "Since the Lower Danube Plain represents one of the oldest areas of continuous human occupation from Neolithic onwards .e., 8000 cal yr BP (http://ran.cimec.ro; BălăÈŹescu and Radu, 2004; Weininger et al., 2009; Wunderlich et al., 2012; Nowacki et al., 2019) and one of the most important agricultural areas in Europe (European Environmental Agency, 2016), the current vegetation is likely very different from its natural state, though exactly how different is not known".

– remove comma after "although". I found this sentence difficult to understand. Are you saying that recent land-cover models tend to support the pollen-based reconstructions, whereas the old ones didn't?

R: Comma removed. Basically yes, the new land cover models tend to better support the pollen-based reconstructions. This sentence now reads: "However, some model-based land cover reconstructions strongly underestimated anthropogenic land cover change in Europe (Gaillard et al., 2010), although recent studies also show a closer match between the model and pollen-based reconstructions of land cover changes (Kaplan et al., 2018). "

– this sentence is very important in the paper, but is very complex and methodological. I feel as though it would be more logical to have it appear after the aim and research questions, rather than before. 96 – these are very good research questions and would benefit from stronger links to the preceding introductory material. Why is it important to know what is "natural" here? Is that a current concern for biodiversity conservation in the region? Why would we expect moisture-demanding taxa in past forests? Has this been hypothesised on some basis? There seems to be a word missing in "When did this area undergo the most marked land cover". . . land-cover change, perhaps?

R: We agree with the reviewer and re-wrote the aims and hypotheses to answer these questions and to ensure a better fit with the themes addressed in the Discussion. See our full replay to his major point 1.

– what is the hypothesis mentioned here? How can it be assessed using pollen data which are poorly taxonomically resolved for many grassland taxa?

R: We did not aim specifically to determine grassland diversity.

– please explain these desertification phenomena earlier in the introduction to provide context

R: We have removed this aim from the paper, therefore no need to address this issue in the Introduction.

– what is the source of the "excessively temperate continental" category? Could you give a Köppen classification as well?

R: See our response to Rev 1_point 2.

– comma after "eroded"

R: Done.

– it's unclear to me why the potential natural vegetation is being presented here, given that the paper aims to reconstruct the past vegetation and land cover. I feel that PNV should be a model to be tested, rather than stated as a fact, as it is here.

R: See our response to the reviewer main point 2.

– the pollen source area is stated as being 20 km. How was this determined?

R: We have determined this by tacking into account the size of the lake, and the number of possible types of habitats/vegetation types.

– what is the minimum pollen sum?

R: Minimum pollen sum was 300 terrestrial pollen grains.

– it would be good to justify the statement that n-alkanes reflect climate conditions here – what is the evidence?

R: Please see our detail response to Rev 1 lines 178, 280)

– this statement about trees may need revision to specify deciduous trees or angiosperms, since n-alkanes appear to be less successful in detecting coniferous trees (Diefendorf et al. 2015. Geochim. Cosmochim. Acta 170: 145-156)

R: We are added a sentence mentioning this bias. See our response to Rev_1 line 178.

220-225 – please provide a reference justifying Zr and Fe:Mn as proxies for erosion and anoxia.

R: We did state this in the methods. This reads: 'We selected detrital element Zr, as proxy for erosion (Kylander et al., 2011) and the Fe: Mn ratio, to reconstruct anoxic conditions in the lake (Nacher et al., 2013)".

– the subtitle here is confusing and unclear. Are you trying to say it's about both periods or the transition from one to the other? Perhaps "Transition from forest steppe (6000-4200 cal. BP) to maximum tree cover (4200-2500 cal. BP)" would be clearer? Also, can we be certain C. orientalis was a 'tree'?

R: See reply to the reviewer's main point 5.

– change "open woodland forest" to "open woodland" R. Changed to open woodlands.

– change "shrub" to "shrubs" (or make grasses and forbs singular)

R: Changed to shrubs.

– change "Coeval to" to "Coeval with"

R: Changed to coeval with. . .

– is it possible to disentangle the climatic and vegetation signals in the n-alkanes, or is it not possible to say whether they were caused by one or the other? Perhaps the pollen record suggests that vegetation thickening was the main cause of the n-alkanes variations? Vegetation thickening might be linked to declining fire activity, changes in agricultural practices (cropping, grazing) and/or climatic drivers.

R: It is not possible with our record of n-alkane to disentangle the climate from vegetation signal. See our response to Rev 1, lines 178, 204.

– how are aquatic plants reflected in the n-alkane record? The absence of aquatic/wetland taxa (e.g. Cyperaceae) in the pollen record (Appendix A1) is striking – were aquatics not counted?

R: We have added and discussed Paq in the alkane record, which quantifies the abundance of submerged and floating vascular macrophytes. We have also added and interpreted the pollen of aquatic and wetland. However, the combined two proxy do not seem to offer a very clear picture of past hydrological changes.

– is it possible that there was a higher lake level when this lake is connected hydrologically to the Danube? This would imply the Danube also having a higher level, wouldn't it? Could the potential anoxia be linked to lake trophic status or to aquatic vegetation?

R: Hydrological connections between Lake Oltina and Danube was likely the case. We have improved this paragraph by bring more evidence for the alternative hypothesis of anoxic based- higher lake level versus trophic status or less turbulent conditions. It reads: The maximum extent of tree cover parallels a slight Fe/Mn ratio increase more evidently between 4000-3500 cal yr BP and 3000-2500 cal yr BP, which may indicate the establishment of more anoxic conditions (Naeher et al., 2013), possibly associated with a higher lake level in connection to the intensification of fluvial activity on Danube river or less turbulent conditions. Increased anoxia linked to a higher lake trophic status and decomposition of organic matter is supported by the slight increase of submerged aquatic macrophyte ( Paq; Fig. 4). The abundance of pollen and spores of aquatic and wetland plants remained rather stable during this time, except for the slight increase aquatic taxa (Potamogeton, Myriophilum, Typa /Sparganim) and a decline in Cyperaceae, at the times of anoxic condition, which may indicate both, a lake level increase as well as a more organic matter in the lake. On the other hand, low values of the lithogenic element Zr between 4200 and 2500 cal yr BP indicate more stable slope conditions with low run off, which might support the hypothesis that increased anoxia may have resulted from less turbulent conditions in the lake.'

– the term "forest cover" is a little bit misleading as the authors' reconstructions do not show the existence of a forest, but of a forest-steppe. There are several instances through the text where this confusion could arise, including line 34 of the Abstract. In the context of forest-steppe, I suggest "tree cover" rather than "forest cover" and perhaps "steppe expansion" rather than "forest loss".

R: We agree with this point and changed forest cover with tree cover throughout the text.

– change "impact of" to "impact on" R: Changed to impact on.

– remove the "." after "Plantago" R. Dot removed.

– the mention of uncertainties is welcome; however, the authors could help the reader to better understand the implications of the uncertainties. . . do the interpretations change if we consider these uncertainties, or are the uncertainties minor?

R: See our detail response to Rev 1. l. 326.

– replace "modelled-based" with "model-based"

R: Changed to model-based.

– unclear sentence

R: We have changed this sentence and now reads: "The comparison of pollen records from the European forest steppe shows a west to east gradient of timing and magnitude of deforestation rate (Fig. 6)".

– change "extend" to "extent"

R: Changed to extent.

– this is an interesting idea about the climatic suitability of the region for forest, though it is maintained relatively open through disturbances like cropping and grazing – a 'plagioclimax'. Perhaps this idea could be expanded a bit more, given its relevance to landscape restoration and conservation, as well to understanding human interactions with the biota of the region.

R: We have slightly expended this at the end of chapter 5.3.

– delete "the" before "SE Europe"

R: Deleted

– comma after "millennia"

R: Inserted.

– the idea that deforestation contributed to aridification would benefit from some discussion and justification in the Discussion section, not just in the Conclusion.

R: We have introduced a short paragraph the end of chapter 5.3.

– there is a slight inconsistency in the argument here about n-alkanes. The authors say that they track the vegetation changes in the pollen, so that makes them a reliable indicator of past vegetation change, but earlier in the paper the n-alkanes are interpreted as a climate proxy. Is it that n-alkanes are a proxy for climate directly, or are they, like pollen, a proxy for vegetation, which can be influenced by climate? Readers like myself will be grateful for the additional explanation!

R: Please see our response to Rev_1 lines 178, 204.

– please help the reader to understand how this single record might be able to be used to test land-cover models – this would add greatly to the application and relevance of the study and 419 – what is meant by "an earlier impact than in the reconstructions is also true"? This issue is not really discussed in the paper and seems not to really relate to the material presented. Perhaps after the research questions are refined, the conclusion could be restructured slightly to address those?

R: Agree. As a comparison between the outcome of pollen and model-based land cover changes was not the main outcome of the paper, we removed this bit from here and the abstract. We have restructured the conclusions to more strongly accommodate our findings.

– replace "design" with "designed"

R: Replaced with designed.

– replace "grating" with "granting" (also line 433)

R: Replaced with granting.

Fig. 5 – please add some indication of the pollen zones from Fig. 4

R: Zones added.

Table 2 – check spelling of Plantago lanceolata

R: Corrected see Table 2.

Appendix A1 – please add wetland/aquatic taxa to the pollen diagrams

R: We have included a pollen of aquatic and wetland into the pollen in the Appendix A1.

---

## Author Response (AR1)

*The manuscript titled "The transformation of the forest steppe in the lower Danube Plain of south-eastern Europe: 6000 years of vegetation and land use dynamic" by Feurdean et al. presents a multi-proxy approach to investigate the history of European forest expansion in the Lower Danube Plain over the last 6000 years. It is a well written and encompasses a detailed comparison to other regions and a comparison to the output of the REVEALS model. Generally, there are no major issues with this manuscript as it is already well organized. I have only minor suggestions and comments which will hopefully help with the data interpretation, particularly of the n-alkane results.*
**R: Thank you for the positive appreciation of our work.**

*Line 68: It would be interesting to know how far in time human occupation in this area has been detected.*
**R: We have revised this paragraph to fit to the story of Potential Natural Vegetation as requested by Rev 2- (see Rev 2 point 2 and lines 92, 96). It now reads:** "Lowland ecosystems (mesic and steppe grasslands, woodlands, etc) provide an array of provisioning (e.g. crops, grazed areas, wood) and regulating services (e.g. soil protection; European Environmental Agency, 2016). However, in comparison to the mountainous areas of central eastern Europe, lowland ecosystems, especially steppe grasslands, have been more strongly impacted by human activities (Magyari et al., 2010…)''. l. 50.

*Line 178: "To determine the climate conditions and the predominant vegetation type. . .". I would suggest to rephrase this sentence and be a little bit more careful with the interpretation value of n-alkanes. n-alkanes are commonly used to distinguish different origins of organic matter sources based on their chain-length and their delta Deuterium values can help to identify past environmental conditions.*
**R: We have rephrased this sentence to more accurately show that the chain-length of n-alkanes are commonly used to distinguish among sources of organic matter. Additionally, to accommodate Rev 2 suggestion (l. 286) on the occurrence of *n*-alkanes of aquatic origin, we have added a new n-alkane ratio, $P_{aq}$ =aquatic index, which quantifies the abundance of submerged vascular macrophyte. This subsection now reads :" 3.2.2 Leaf wax *n*-alkanes based vegetation reconstruction**. To determine the source of organic matter and the predominant vegetation type (Eglinton and Calvin, 1967; Ficken et al., 2000; Diefendorf et al. 2015), we measured the concentration of higher-plant derived *n*-alkane homologues of 60 sediment samples selected along the composite core. *n*-Alkanes are an integral part of higher-plant leaf epicuticular waxes, highly resistant to degradation and among the most stable lipid components of the protective waxes coating terrestrial plant leaves (Eglinton and Eglinton, 2008; Sachse et al., 2012). They are commonly used to distinguish sources of organic matter based on their chain-length (see below). In this study, we calculated the ratio of straight-chain *n*-alkanes of different chain lengths (homologues) as these have been previously used as proxies for the relative contribution of various types of plants in lacustrine sediments (e.g. Ficken et al., 2000; Zhou et al., 2005). Average chain length (ACL) is an indicator of the relative abundance of short ($C_{16}$-$C_{20}$) vs long chain *n*-alkanes and may be linked to the predominance of higher taxonomic plants over lower taxonomic plants (Ficken et al., 2000; Eglinton and Calvin, 1967). Within the long chain *n*-alkanes, the abundance of *n*-alkanes with *n*-$C_{31}$ and *n*-$C_{33}$ may be indicative of grass predominance, whereas *n*-$C_{27}$ and *n*-$C_{29}$ may indicate a predominantly tree covered landscape (Aichner et al., 2010; Meyers, 2003). The aquatic index ($P_{aq}$) quantifies the abundance of submerged and floating vascular macrophytes, which are characterised by medium chain length *n*-alkanes, relative to emergent plant types that are characterised by long chain *n*-alkanes (Ficken et al., 2000). It should be noted that *n*-alkanes are less successful in detecting coniferous than angiosperms (Diefendorf et al. 2015) and that some overlap within the mid chain length alkanes *n*-$C_{23}$ and *n*-$C_{25}$ is possible (Aichner et al., 2010; Meyers, 2003). L.205-210.

*Line 183: N2 subscript Chapter 3.3 Regime disturbances by fire and herbivores: Since you aim to reconstruct past fires, have you considered analyzing levoglucosan as a proxy for biomass burning?*
**R: We could not reliably detect levoglucosan in any of the trial samples. Levoglucosan peaks detected in a few samples were very small relative to the high background, meaning a high degree of uncertainty of the result.**

*Line 280: Please explain how shorter chain lengths of n-alkanes correlate to moister climate conditions. Line*

**R: To avoid confusion (see also comment l.282 of Rev 2), we have removed the interpretation of the shorter chain length as a direct proxy for moist conditions and retained only that the $(C_{27}+C_{29})/(C_{33}/C_{31})$ ratio primary reflect changes in major vegetation type. However, indirectly, increases in the $(C_{27}+C_{29})/(C_{33}/C_{31})$ ratio typical for the dominance of trees may indicate wetter conditions, as trees required higher amount of moisture than the grasslands. This sentence now reads:** "Coeval with the maximum extent in tree cover, the $n$-alkanes were dominated by the shorter chain lengths (ALC) and a higher $n(C_{27}+C_{29})/(C_{33}/C_{31})$ ratio, indicative of an increased contribution of tree-derived $n$-alkanes (Meyers, 2003; Aichner et al., 2010). However, the concentration of individual $n$-alkanes $>C_{27}$ (not shown) varied with that of detrital element Zr. It was high between 6000 and 4200 cal yr BP and decreased between 4200 and 2500 cal yr BP, suggesting a reduction in the terrestrial plant delivery into the lake during the highest tree cover" l.305.

*2. Study area. 290: Since you discuss climate dynamics in your study region, you should add a short paragraph about the current climate regime/atmospheric systems effecting the studied lake in section*
**R: This and the suggestions of Rev2_114 we have revised the information about the climate type in this region. This reads: "**The climate in the study region is wet-warm temperate continental (Koppen-Geigger class Dfa), also termed excessive with the prevalence of harsh winters and hot summers (Posea et al., 2005), due to the influence of air masses from continental Asia". l. 120.

*Line 317: "As climate conditions remained relatively moist during this decline, anthropogenic rather than climatic causes are likely" How did you infer moist conditions?*
**R: We have removed the climatic hypothesis from this place, and only discussed pollen-based evidence of anthropogenic land cover changes. In the second part of this subchapter (5.2), we discuss local climate conditions based on P$_{aq}$, and geochemistry as well as published regional climatic records and conclude that climate conditions remained relatively moist during this decline in tree cover, therefore anthropogenic impact was likely the main cause of tree cover loss.** l. 320 to 340.

*Line 326: "The proportion of Cerealia is significantly greater in the REVEALS estimate (40%) than in the raw pollen percentages (5%)" Interesting, why is that?*
**R: To accommodate this comment and at the suggestion of Rev2 (line 329) we have introduced a paragraph discussing the effect of uncertainties in the REVEALS model on the pollen-based vegetation reconstruction. This reads: "**Our Cerealia pollen includes *Triticum, Zea, Hordeum* and *Secale cereale*, for which we have used productivity estimates derived from the calibration of local surface pollen samples with a vegetation inventory (Grindean et al., 2019). These productivity estimates are considerably lower than the average for Europe (0.22 vs. 1.85; Mazier et al., 2012) and therefore the main cause of the high proportion of Cerealia cover reconstructed by REVEALS and for the disparity between the outcome of this study and elsewhere in Europe. Furthermore, the crop species included in Cerealia also vary regionally and with time, which may also introduce variation when applying their PPEs for landscape reconstruction over an extended period of time. Lastly, the occurrence of wild grass species with pollen that fall in the Cerealia pollen type (all Poaceae grains larger than 40 μm), may have led to an overestimation of the proportion of Cerealia at certain times in the past" l. 360-365.

*Line 367: "Our quantitative record of vegetation indicates a higher than present tree cover across the landscape of the eastern Lower Danube Plain between 6000 and 4200 cal yr BP with an absolute maxima of 50% (60% raw pollen percentages) between 4200 and 2500 cal yr BP (Fig. 6, see Table 3 for reference and Fig. 1 for locations)." Does the concentration of n-alkanes also increased between 6000 and 4200 cal yr BP?*
**R: We have extended the discussion on the *n*-alkane concentration.** See l. 395 and our reply l. 280.

*You compare your results to a various other locations. However, did you also consider altitude variations?*
**R: All sites are located at low elevation (lower than 300 m asl). To illustrate this, in Table 3, we have added the elevation of each site.**

*Figures Figure 3: I would suggest to delete the number 850 and possibly 950 from the y axis of the depth [cm] since they are too close to the previous and subsequent depth numbers and thus nearly illegible.*
**R: Done, see new figure 3.**

Figure 4: The numbers on the x axis and y axis are a little bit hard to read due to the small font size.
**R: Done, see new figure 4.**

*BGD Interactive comment* Biogeosciences Discuss., *https://doi.org/10.5194/bg-2020-239-RC2, 2020© Author(s) 2020. This work is distributed under the Creative Commons Attribution 4.0 License. Interactive comment on "The transformation of the forest steppe in the lower Danube Plain of south-eastern Europe: 6000 years of vegetation and land use dynamic" by Angelica Feurdean et al. Simon Connor (Referee)* simon.connor@anu.edu.au

*This manuscript, by Angelica Feurdean and colleagues, is an interesting and professionally executed study of past conditions on the Lower Danube Plain. The authors use a multiproxy approach and quantitative land-cover modelling to address questions about the past extent and dynamics of the forest-steppe ecotone in the Western Black Sea region. The manuscript is very well referenced, contains high quality data and is clearly written. The high temporal resolution of the sampling and the quantitative modelling aspects really make this paper stand out from all others in the region. I agree with the other reviewer (#1) that the manuscript presents no major problems. However, I do have a number of suggestions for further improvement that may improve the manuscript's structure and its interdisciplinary and international appeal.*

**R: Many thanks for your positive appreciation of our work.**

Specific comments:

*1.The Introduction would benefit from some careful restructuring to link the literature review to the research questions more closely. At the moment, the Introduction seems to be presenting many different aims and objectives: "determine lake catchment and in-lake changes", "explore the role of climate, natural and anthropogenic disturbances", test a hypothesis about the naturalness of the landscape, determine whether forests were more moisture-demanding, determine the timing of transformations, determine the ecosystem's sensitivity to climate and anthropogenic impacts, inform decision making about desertification and to test land-cover models. This seems like too many questions for one paper – the authors cannot hope to deliver on all of these with depth and certainty. Indeed, many of these themes are not revisited in the Discussion section. A more focussed introduction would clarify exactly what problems the authors are aiming to (and can) solve. Ideally, the research questions should arise from gaps or uncertainties in the literature.*

**R: As this study is the first mid to late Holocene multi-proxy palaeoenvironmental and palaeovegetation reconstruction for this region, we highlighted all the many questions we felt we could answer with the current datasets. In the revised version of the paper, we streamlined the aims to ensure a better fit with the themes addressed in the Discussion**, i.e., understanding the past vegetation composition and dynamics and their drivers (climate, humans and fire). The aims now read: "Here, we explore the long-term vegetation dynamics of the Lower Danube Plain landscapes and the competing driving forces (climate, fire and anthropogenic impact). More specifically, we address the following research questions:

   i)     Is forest steppe the natural vegetation type of the Lower Danube Plain under climatic conditions similar to those in the present?
   ii)    Has the tree cover been more extensive or dominated by other tree taxa in the past?
   iii)   When did this area undergo the most marked land cover and land use changes and was this transformation continuous in time?

This study is built on a pollen-based quantitative vegetation reconstruction (REVEALS model), along with records of long-chain higher plant-wax *n*-alkanes, charcoal, coprophilous fungi and geochemistry from a sedimentary sequence from Lake Oltina, south-eastern Romania. This is the first pollen-based quantitative land cover estimate in the forest steppe of south eastern Europe. It allows the hypothesis of the naturalness of forest steppe ecosystems in this region, as well as its sensitivity to climate and anthropogenic impact, to be critically tested. l.100-110.

*2.The authors' use of Potential Natural Vegetation (PNV) as a baseline could be more critically assessed. PNVs are problematic since they are static in space and time, while pollen data and REVEALS reconstructions, like the results presented here, show that the past vegetation was spatio-temporally dynamic. There is an excellent paper exploring the mismatches between PNV and REVEALS in Czechia (Abraham et al. 2016, Preslia 88: 409-434) and I encourage the authors to consult it and other papers on the topic (e.g. Chiarucci et al. 2010, J. Veg. Sci. 21: 1172-1178; Rull 2015, J. Veg. Sci. 26: 603-607). My feeling is that the manuscript would be stronger if the authors reduced their reliance on the PNV map and instead used the REVEALS reconstruction as a test for PNV accuracy. This is important because PNV is commonly used as a baseline for restoration and it could be influencing current conservation efforts. The paper shows that there are several possibilities for the vegetation of the site – various types of forest-steppes, steppes and agro-pastoral landscapes, which are dynamic in space and time. This would make for much richer and more interesting conclusions.*

**R: This is a pertinent point. As this point appeared to not have come across clearly, in the revised version, we have incorporated the reviewer's suggestions and more specifically highlight the issues of using PNV as**

**a baseline for vegetation composition. This now reads:** ''According to Bohn et al. (2003), the potential natural vegetation (PNV) in the study area, the easternmost part of the Lower Danube Plain, also known as the Southern Dobrogea Plateau, is forest steppe, i.e., woodland patches within a matrix of graminoid and forb dominated communities. It borders steppe grasslands i.e., treeless vegetation dominated by graminoids and forbs to the east. The forest steppe and steppe vegetation extend over 6000 km along an east-west gradient across Eurasia (Fig. 1) under climate conditions delimited by a ca. 2 month long late-summer drought for the forest steppe zone and 4–6 months for the steppe (Walter, 1974). However, according to the management plan for the region (Planul de Management, 2016), there are currently few patches of natural steppe vegetation preserved in the Southern Dobrogea Plateau as most have become ruderal steppe. Since the Lower Danube Plain represents one of the oldest areas of continuous human occupation from Neolithic onwards i.e., 8000 cal yr BP (http://ran.cimec.ro; Bălășescu and Radu, 2004; Weininger et al., 2009; Wunderlich et al., 2012; Nowacki et al., 2019: Preoteasa et al., 2019) and one of the most important agricultural areas in Europe (European Environmental Agency, 2016), the current vegetation is likely very different from its natural state, though exactly how different is not known. The use of PNV as a baseline for natural vegetation has been found to be problematic as it considers only climax vegetation i.e., the final stage of an ecological succession, and assumes that vegetation remains static in space and time (Chiarruci, et al., 2010; Jackson et al., 2013; Abraham et al., 2016; Rull 2015). PNV also fails to consider the vital role of natural disturbances such as fire and herbivores and their impacts on vegetation succession (Chiarruci, et al., 2010; Feurdean et al., 2018), leading to unrealistic vegetation reconstruction even in the absence of disturbance by humans (Jackson et al., 2013). As a consequence, many areas that are currently covered by grasslands or open woodlands in Central Eastern Europe are defined as naturally dominated by deciduous broadleaf forest or mixed coniferous and broadleaf forest as the PNV (Feurdean et al., 2018) and the representation of pioneer trees is much lower compared to pollen-based Holocene vegetation estimates (Abraham et al., 2016). Inaccurate identification of natural vegetation types can also lead to inappropriate decision making in terms of conservation practices and policies. For example, the Global Partnership on Forest and Landscape Restoration use PNV to identify opportunities for landscape restoration to mitigate climate change in areas where the climate can sustain forest (http://www.wri.org/applications/maps/flr-atlas). However, this approach threatens grassland ecosystems as such policies are based on the false assumption that most grasslands are man-made and ignores the very high richness of grasslands at smaller spatial scale (Whittaker et al. 2001), as well as their unique cultural significance (Dengler et al., 2014). **.''** l. 55-80.

*3.Given the carbonate-rich geology and the fact that mostly shells were dated, it would be useful to briefly discuss potential reservoir effects. Reservoir effects will not change the modelling results, but may introduce some uncertainty about the timing of the major transitions identified in the land-cover reconstructions and the interpretations of vegetation change as being forced by climate*

**R: We have determined that the hard water effect is about 1000 years and discuss the timing of major environmental changes assuming ages 1000 years younger than would have been calculated without taking into account this dating limitation (see Appendix 1). In the revised discussion we also discuss the timing of major deforestation and the link with archaeological evidences and simulated vegetation using both original and adjusted age.** See l. 390.

*4. A surprising omission in the manuscript is the aquatic pollen taxa. These taxa might help better interpret the n-alkane results, providing additional proxy for lake hydrological conditions. The authors' interpretations of anoxia and lake level change may find stronger support with the addition of aquatic pollen. The extent to which aquatic vegetation influenced the n-alkanes signal could also be explored.*

**R: We have included aquatic and wetland taxa into the full pollen diagram (Appendix A2) and discuss potential connection with hydrological conditions in the revised version of the paper.** See l. 315

*5. The REVEALS modelling in this paper is very comprehensive. One small doubt concerns Carpinus orientalis. In my experience, this 'tree' is very often a shrub in forest-steppe landscapes. Is it correct to call it a 'forest' taxon in this region? Is it possible that the increase in C. orientalis around 4200 cal. yr BP represents a scrub expansion or the abandonment of coppicing? These changes may even relate to shifting land-use at the Neolithic–Bronze Age transition, a topic that would benefit from further exploration in the Discussion.*

**R: According to the European Atlas of Forest Tree Species (2016), *Carpinus orientalis* is described as a small tree or shrub, and as a tree species native to southeast Europe, the Pontic region and western Asia. In Bulgaria, Dodev et al. (2020) Austrian Journal of Forests Science, 137, 23-42, 2020, defined oriental hornbeam forests as forest stands with 50 % or more of *C. orientalis* in the tree composition. Therefore, we have referred to *C. orientalis* as a forest taxon based on its percentages and consistent presence in the uppermost canopy levels in our vegetation surveys of/within forest stands in the area. The increase of *Carpinus orientalis* around 4200 cal BP at the expense of *Quercus* is concurrent with the expansion of *Fagus***

*sylvatica* and *C. betulus* and a low abundance of other local shrub taxa (Appendix A2), thus suggesting its expansion as a tree not scrub expansion. The decrease of *Quercus* could indicate deforestation, although the low abundance of anthropogenic pollen indicators (Cerealia) and coprophilous fungi suggest logging was at a small scale, probably for individual households.

Technical comments by line number:

*31 – place a comma after "taxa" 32 – "Maximum tree cover. . . between 4200 and 2500" here, but "greatest tree cover. . . between 6000 and 2500" in line 30. . . this seems like a mistake since the ranges overlap*
**R: Done, this sentence enow reads: "We found a greater tree cover, composed of xerothermic (*Carpinus orientalis* and *Quercus*) and temperate (*Carpinus betulus*, *Tilia, Ulmus* and *Fraxinus*) tree taxa, between 6000 and 2500 cal yr BP. Maximum tree cover (~50%) occurred between 4200 and 2500 cal yr BP at a time of wetter climatic conditions.** l. 30.

*35 – "mid-Holocene forest maximum" seems to refer to conditions from 4200-2500 cal. yr BP, which is usually termed late Holocene.* Line 34-See also comments about the use of "forest", line 305. In the context of forest-steppe, I suggest "tree cover" rather than "forest cover" and perhaps "steppe expansion" rather than "forest loss".
**R: We have corrected mid with late Holocene. Additionally, we have changed forest cover with tree cover through the manuscript.**

*38 – "highlighting recurring anthropogenic pressure" – is this indeed cyclical, or due to more-or-less continuous anthropogenic pressure?*
R. **Changed.** ''A sparse tree cover, with only weak signs of forest recovery, then became a permanent characteristic of the Lower Danube Plain highlighting more or less continuous anthropogenic pressure''. L.35.

*39 – "was in between that in" – consider "falls in between that of"*
**R: Changed to: "The timing of anthropogenic ecosystem transformation here (2500 cal yr BP) falls in between that in central eastern (between 3700 and 3000 cal yr BP) and eastern (after 2000 cal yr BP) Europe".** l.40.

*42 – "reflects" – change to "reflect" to agree with woodlands*
**R: Changed.**

*44 – the comment about pollen preservation seems to indicate that this pollen record may be adversely affected by taphonomy. . . is that the case? Perhaps taphonomy could be addressed elsewhere in the paper?*
**R: We did not run into the problem of pollen preservation at Lake Oltina. We remove this sentence from the abstract and preserve it at the conclusion to highlight that *n-alkanes* could accurately track changes in the main vegetation types i.e., tree-grass cover in this region.**

*47 – delete "The" at start of sentence 49-51 – this sentence sounds like it concerns the present-day environment (i.e. ecological studies), but all the references are palaeo studies. More precise wording would avoid misleading readers, or modern ecological studies could be incorporated to increase the interdisciplinarity of the manuscript 54 – the definition of "lowlands" in this paper is not very clear, since it seems to imply that lowlands are "drier" than "mesic areas of Europe". A more precise term like "steppic grasslands" or even a map showing the current extent (not potential extent) of the ecosystem might help, since readers could easily confuse the steppic lowlands they are referring to with the lowlands of the Netherlands, for example, which are certainly quite different.*
R: We have split this sentence in two; the first refers to contemporary studies the second to past trends. It reads:
*"Lowland ecosystems (mesic and steppic grasslands, woodlands, etc) provide an array of provisioning (e.g. crops, grazed areas, wood) and regulating services (e.g. soil protection; European Environmental Agency, 2016). However, in comparison to the mountainous areas of central eastern Europe, lowland ecosystems, especially steppe grasslands, have been more strongly impacted by human activities* (Magyari et al., 2010…)*".* l.50.

*65 – the idea of grasslands being richer than rainforests seemed surprising, but the original reference states "at small spatial scales vascular plant diversity of certain European grasslands even exceeds tropical rainforests". It would be good to specify what scale is meant here, since 'alpha diversity' is a vague term encompassing a large range of spatial scales (Whittaker et al. 2001, J. Biogeog. 28: 453-470). 67 – this sentence refers to temperate grasslands (it may be different for tropical grasslands)*
**R: Added. It reads:"** However, this approach threatens grassland ecosystems as such policies are based on the false assumption that most grasslands are man-made and ignores the very high richness of grasslands at smaller spatial scale (Whittaker et al. 2001), as well as their unique cultural significance (Dengler et al., 2014). l..80.

*71 – unclear what you mean by "cultural rise" – what is it and what is the evidence for it?*

**R: We have revised this paragraph. It reads: "**Since the Lower Danube Plain represents one of the oldest areas of continuous human occupation from Neolithic onwards i.e., 8000 cal yr BP (http://ran.cimec.ro; Bălăşescu and Radu, 2004; Weininger et al., 2009; Wunderlich et al., 2012; Nowacki et al., 2019: Preoteasa et al., 2019) and one of the most important agricultural areas in Europe (European Environmental Agency, 2016), the current vegetation is likely very different from its natural state, though exactly how different is not known*".* l.65.

*87 – remove comma after "although". I found this sentence difficult to understand. Are you saying that recent land-cover models tend to support the pollen-based reconstructions, whereas the old ones didn't?*

**R: Basically yes, the new land cover models tend to better support the pollen-based reconstructions. This sentence now reads: "**In addition, models of deforestation rates, using a scenario that accounts for population history and technological advances, suggest that the extent of deforestation in the lower Danube basin has increased continuously since 4000 cal yr BP (Kaplan et al., 2009; Giosan et al., 2012). l. 95.

*92 – this sentence is very important in the paper, but is very complex and methodological. I feel as though it would be more logical to have it appear after the aim and research questions, rather than before.*
*96 – these are very good research questions and would benefit from stronger links to the preceding introductory material. Why is it important to know what is "natural" here? Is that a current concern for biodiversity conservation in the region? Why would we expect moisture-demanding taxa in past forests? Has this been hypothesised on some basis? There seems to be a word missing in "When did this area undergo the most marked land cover". . . land-cover change, perhaps?*

**R: We agree with the reviewer and re-wrote the aims and hypotheses to answer these questions and to ensure a better fit with the themes addressed in the Discussion. See our full reply to his major point 1.**

*103 – what is the hypothesis mentioned here? How can it be assessed using pollen data which are poorly taxonomically resolved for many grassland taxa?*
**R: We did not aim specifically to determine grassland diversity.**

*105 – please explain these desertification phenomena earlier in the introduction to provide context*
**R: We have removed this aim from the paper, therefore no need to address this issue in the Introduction.**

*114 – what is the source of the "excessively temperate continental" category? Could you give a Köppen classification as well?*
**R: See our response to Rev 1_point 2.**

*118 – comma after "eroded"*
**R: Done.**

*120 – it's unclear to me why the potential natural vegetation is being presented here, given that the paper aims to reconstruct the past vegetation and land cover. I feel that PNV should be a model to be tested, rather than stated as a fact, as it is here.*
**R: See our response to the reviewer main point 2.**

*126 – the pollen source area is stated as being 20 km. How was this determined?*
**R: We have determined this by taking into account the size of the lake, and the habitats/vegetation types surrounding the lake.**

*162 – what is the minimum pollen sum?*
**R:  Minimum pollen sum was 300 terrestrial pollen grains**. l. 170.

*177 – it would be good to justify the statement that n-alkanes reflect climate conditions here – what is the evidence?*
**R: Please see our detail response to Rev 1 lines 178, 280.**

*199 – this statement about trees may need revision to specify deciduous trees or angiosperms, since n-alkanes appear to be less successful in detecting coniferous trees (Diefendorf et al. 2015. Geochim. Cosmochim. Acta 170: 145-156)*
**R: We are added a sentence mentioning this bias. See our response to Rev_1 l. 178.**

*220-225 – please provide a reference justifying Zr and Fe:Mn as proxies for erosion and anoxia.*
**R: We stated this in the methods. This reads: '**We selected detrital element Zr as proxy for erosion (Kylander et al., 2011) and employed the Fe:Mn ratio to reconstruct anoxic conditions in the lake (Nacher et al., 2013).**''.** l. 155.

*273 – the subtitle here is confusing and unclear. Are you trying to say it's about both periods or the transition from one to the other? Perhaps "Transition from forest steppe (6000-4200 cal. BP) to maximum tree cover (4200-2500 cal. BP)" would be clearer? Also, can we be certain C. orientalis was a 'tree'?*
**R: See our reply to the reviewer's point 5.**

*276 – change "open woodland forest" to "open woodland"*
**R. Changed to open woodlands.**

*279 – change "shrub" to "shrubs" (or make grasses and forbs singular)*
**R: Changed to shrubs.**

*280 – change "Coeval to" to "Coeval with"*
**R: Changed to coeval with…**

*282 – is it possible to disentangle the climatic and vegetation signals in the n-alkanes, or is it not possible to say whether they were caused by one or the other? Perhaps the pollen record suggests that vegetation thickening was the main cause of the n-alkanes variations? Vegetation thickening might be linked to declining fire activity, changes in agricultural practices (cropping, grazing) and/or climatic drivers.*
**R: It is not possible with our record of *n*-alkanes to disentangle the climate from vegetation signal. See our response to Rev** 1, l. 178, 204.

*286 – how are aquatic plants reflected in the n-alkane record? The absence of aquatic/wetland taxa (e.g. Cyperaceae) in the pollen record (Appendix A1) is striking – were aquatics not counted?*
**R: We have added and discussed $P_{aq}$ in the n-alkanes record (Fig. 3), which quantifies the abundance of submerged and floating vascular macrophytes, as well as pollen of aquatic and wetland in Apendix A2.** See l. 315, 395.

*288 – is it possible that there was a higher lake level when this lake is connected hydrologically to the Danube? This would imply the Danube also having a higher level, wouldn't it? Could the potential anoxia be linked to lake trophic status or to aquatic vegetation?*
**R: Hydrological connections between Lake Oltina and Danube was likely the case. We have improved this paragraph by bring more evidence for the alternative hypothesis of anoxic based- higher lake level versus trophic status or less turbulent conditions. It reads: ''**The maximum extent of tree cover parallels a slight Fe:Mn ratio increase more evidently between 4000-3500 cal yr BP and 3000-2500 cal yr BP. This may indicate the establishment of more anoxic conditions (Naeher et al., 2013), possibly associated with a higher lake level due to the intensification of Danube water and sediment discharge, a higher lake trophic status, or less turbulent conditions. Increased anoxia linked to a higher lake trophic status and the decomposition of organic matter is supported by the slight increase of submerged aquatic macrophyte ($P_{aq;}$ Fig. 4). The presence of more aquatic plants in the lake at the time of the rise in Fe:Mn ratio is also suggested by the increased abundance of aquatic/wetland taxa (*Potamogeton*, *Myriophilum*, *Typa /Sparganim*). On the other hand, low values of the lithogenic element Zr between 4200 and 2500 cal yr BP indicate more stable slope conditions with low run off, which might support the hypothesis that increased anoxia may have resulted from less turbulent conditions in the lake''. l. 305 to 315.

*305 – the term "forest cover" is a little bit misleading as the authors' reconstructions do not show the existence of a forest, but of a forest-steppe. There are several instances through the text where this confusion could arise, including line 34 of the Abstract. In the context of forest-steppe, I suggest "tree cover" rather than "forest cover" and perhaps "steppe expansion" rather than "forest loss".*
**R: We agree with this point and changed forest cover with tree cover throughout the text.**

*311 – change "impact of" to "impact on"*
**R: Changed to impact on.**

*325 – remove the "." after "Plantago"*

**R. Dot removed.**

*329 – the mention of uncertainties is welcome; however, the authors could help the reader to better understand the implications of the uncertainties. . . do the interpretations change if we consider these uncertainties, or are the uncertainties minor?*
**R: See our response to Rev 1.** l. 326.

*353 – replace "modelled-based" with "model-based"*
**R: Changed to model-based.**

*392 – **unclear sentence***
**R: We have changed this sentence to:** *"The comparison of pollen records from the European forest steppe shows a west to east gradient in the timing and magnitude of deforestation (Fig. 6)"*. l. 430.

*398 – change "extend" to "extent"*
**R: Changed to extent.**

*399 – this is an interesting idea about the climatic suitability of the region for forest, though it is maintained relatively open through disturbances like cropping and grazing – a 'plagioclimax'. Perhaps this idea could be expanded a bit more, given its relevance to landscape restoration and conservation, as well to understanding human interactions with the biota of the region.*
**R: For now we have expended the text to show the difference between pollen based and map based vegetation as well implication of desertification see** l .435.l 440 and l.460 (conclusion)**.**

*403 – delete "the" before "SE Europe"*
**R: Deleted**

*411 – comma after "millennia"*
**R: Inserted.**

*412 – the idea that deforestation contributed to aridification would benefit from some discussion and justification in the Discussion section, not just in the Conclusion.*
**R: We have introduced a short paragraph the end of chapter 5.3.** see l .440.

*416 – there is a slight inconsistency in the argument here about n-alkanes. The authors say that they track the vegetation changes in the pollen, so that makes them a reliable indicator of past vegetation change, but earlier in the paper the n-alkanes are interpreted as a climate proxy. Is it that n-alkanes are a proxy for climate directly, or are they, like pollen, a proxy for vegetation, which can be influenced by climate? Readers like myself will be grateful for the additional explanation!*
**R: Please see our response to Rev_1,** l. 178, 204.

*418 – please help the reader to understand how this single record might be able to be used to test land-cover models – this would add greatly to the application and relevance of the study and 419 – what is meant by "an earlier impact than in the reconstructions is also true"? This issue is not really discussed in the paper and seems not to really relate to the material presented. Perhaps after the research questions are refined, the conclusion could be restructured slightly to address those?*
**R: Agree. As a comparison between the outcome of pollen and model-based land cover changes was not the main outcome of the paper, we remove this bit from here and the abstract. We have restructured the conclusions to more strongly accommodate our findings.**

*425 – replace "design" with "designed"*
**R: Replaced with designed.**

*431 – replace "grating" with "granting" (also line 433)*
**R: Replaced with granting.**

*Fig. 5 – please add some indication of the pollen zones from Fig. 4*

**R: Zones added.**

*Table 2 – check spelling of Plantago lanceolata*
**R: Corrected see Table 2.**

*Appendix A1 – please add wetland/aquatic taxa to the pollen diagrams*
**R: We have included a pollen of aquatic and wetland taxa into the pollen in the Appendix A2.**

Your sincerely,
Angelica Feurdean